Subject Areas:
geology/geophysics/petrology

Keywords:
diagenesis, water saturation, conductivity characteristics, rock core experiments, electrical properties

Author for correspondence:
Zhansong Zhang
e-mail: zhangzhs@yangtzeu.edu.cn

# Comparative study and discussion of diagenetic facies and conductivity characteristics based on experiments

Xueqing Zhou[1,2], Zhansong Zhang[2] and Chaomo Zhang[2]

[1]Institute of Deep-sea Science and Engineering, Chinese Academy of Sciences, Sanya 572000, People's Republic of China
[2]Key Laboratory of Exploration Technologies for Oil and Gas Resources, Ministry of Education, Wuhan 430100, People's Republic of China

ZZ, 0000-0003-1348-7228

The study of conduction mechanisms is the key to establishing physical derivations, resistivity simulations and saturation models. The purpose of this research is to clarify conduction mechanisms under different diagenetic facies and build suitable saturation evaluation models. Experimental data of tight gas sandstone from the Ordos Basin were analysed, including data from scanning electron microscopy, conventional core physical property analysis, core casting thin-section analysis, core mercury intrusion experimentation and rock electrical conductivity experimentation. Accordingly, the diagenetic minerals of the study block were examined, and the diagenetic facies were classified by the differences in the diagenetic properties across the study area. The reservoir was divided into three types of diagenetic facies: construction facies, cementation facies and destruction facies. On this basis, the conductivity characteristics and saturation models of different diagenetic facies within the study area were systematically discussed for the first time. A number of experiments showed that according to the type of diagenesis, the structure of the pores and the influence of the reservoir, a classification scheme for diagenetic facies (consisting of construction, cementation and destruction facies) can be established. According to the influence of the diagenesis of various diagenetic facies, theoretical pore structure models of the three diagenetic facies were established, in which the construction facies includes mainly dissolved feldspar pores and intergranular pores, the destruction facies includes clay residual intergranular pores and intergranular pores, and the cementation facies includes primarily residual intergranular pores. Based on these theoretical pore structure models, the

construction facies was evaluated with a pore-connected vuggy conductivity model, the destruction facies was evaluated with a non-connected matrix pore conductivity model, and the cementation facies was evaluated with a residual intergranular pore conductivity model. Then, the rationality of each model and the effects of the parameters in each model on the final cementation exponent were analysed by simulation. The predicted cementation exponents of the diagenetic facies match the measured cementation exponents well and can guide the qualitative description of these characteristics in such reservoirs.

## 1. Introduction

Tight gas sandstone reservoirs contain unconventional resources that have been successfully developed around the world [1]. Because tight sandstone reservoirs have experienced relatively strong diagenesis [2], they are generally characterized by variable lithologies, complex pore structures, diverse pore types, highly developed secondary pores, large pore size differences and strong heterogeneity [3]. The conductivity of a logging curve is complex [4], resulting in tight sandstones that exhibit 'non-Archie' behaviour [5]. To study the complex conductivity properties of tight sandstone reservoirs, scholars have evaluated the saturation of low-permeability sandstone reservoirs [6]. Two main ideas have emerged from this research. (i) Saturation models are based on a single controlling factor [7–9]. This principle originates mostly from analysing the factors that influence the rock electrical parameters, determining the dominant factors governing the conductive characteristics of rock, and establishing the corresponding conductive model [10,11], including physical experiments and numerical simulations revealing the influence of single factors on the electrical conductivity of rocks in the reservoir [12]. Many scholars have studied the influence of factors such as shale content, pore structure and pore type on rock electrical parameters, and the effects of individual factors (such as the shale content, pore structure and pore type) on rock conductivity have been discussed in depth. However, the rock electrical parameters of tight sandstone reservoirs are usually controlled by multiple factors [13]; therefore, establishing a reasonable interpretation model by analysing only individual factors is difficult.

(ii) Saturation models are based on classifications [14,15]. For complex reservoirs with variable lithologies, complex pore structures, diverse pore types, secondary porosity, large pore size differences and considerable heterogeneity, the key is to investigate the fundamental cause of the complexity [16]. Several methods have been developed to subdivide the classification units of sandstone reservoirs; these methods are summarized into two types. The first type of classification method is an empirical classification technique based on three categories of petrophysical differences: (i) pore structure parameters [17–19], (ii) hydraulic flow units [20–23], and (iii) functional classification [24,25]. The other type of classification method is based on differences in geological origin [26], including lithofacies, sedimentary microfacies and diagenetic facies. Among them, diagenesis is one of the most important factors influencing the reservoir quality of sandstones (sedimentary environments is another), and diagenetic facies describe the comprehensive characteristics of present-day reservoirs by reflecting the properties and types of reservoirs. Research on diagenetic facies is of broad significance for predicting and evaluating the presence of high-quality reservoirs [27,28], the controls of certain physical parameters [29] and the effects of certain processes on microscopic pore throat structures and pore types. Certain achievements have been reported from other perspectives on this subject. Therefore, diagenetic facies may be a suitable geological unit for highly accurate saturation calculations.

To represent reservoirs with complicated pore structures or rock properties strongly influenced by diagenesis, which lead to complex conductivity characteristics, a tight gas sandstone reservoir in the LX block in the northwestern Ordos Basin is taken as an example. This study focuses on the analysis of the conduction mechanism and the establishment of a conductivity model corresponding to different diagenetic facies characteristics. First, on the basis of data from comprehensive core casting thin-section analysis, rock physical property analysis, rock electrical conductivity experiments, and logging, the diagenetic facies in the LX block are classified. Second, the mechanisms responsible for controlling the formation and conductivity of diagenetic facies are studied. Then, the saturations of different diagenetic facies are established by theoretical derivation. The resulting model is based on a thorough understanding of the formation mechanisms and conductivity characteristics of reservoirs and can assist in evaluating the saturation of a complex reservoir that has experienced strong diagenesis.

## 2. Geological background

### 2.1. Basin evolution

The Ordos Basin, which is a relatively simple yet large cratonic sedimentary basin, is located on the western edge of the North China Plate. The perimeter of the basin is rectangular. According to its structure and evolution, the Ordos Basin is divided into six first-order tectonic units [30]. The study area is set within a structural unit of the generally monoclinic Jinxi flexure, and the central eastern region is an uplifted zone affected by the tectonic activity of the Zijinshan mineral field. The regional topography of the LX block, which is influenced by tectonic evolution in the Ordos Basin, is high in the east and low in the west. The two main factors that led to the formation of tight sandstones in the study area with unique geological features are sedimentation and diagenesis [31,32]. The resulting reservoir has the characteristics of low porosity and permeability (less than 0.1 mD).

### 2.2. Stratigraphy and depositional facies

In the upper Permian layers of the Ordos Basin, sedimentary microfacies, such as subaerial distributary channel microfacies, subaqueous distributary channel microfacies and estuary bar microfacies, are mainly developed, and most oil and gas productivity occurs within these three main sedimentary microfacies. In the lower Permian layers, tidal flat and sand flat sedimentary microfacies are developed. Distributary channel deposits represent the underwater extensions of distributary channels in a delta plain. Here, sand silt is the main type of sediment, and minimal shale is present. The underwater distributary channel deposits are generally dominated by sandstone and conglomerate-bearing sandstone; the conglomerate is relatively thin and fine-grained, and its sequence is characterized by a scouring surface at the bottom. An estuary dam is located in the estuary of the underwater distributary channels. Horizontal bedding and wavy bedding are present in the lower part of the channel, and parallel bedding appears at the top. In addition, slump deformation bedding and drainage structures are well developed. Tidal flat and sand flat sedimentary microfacies are mostly developed near the coastline and are divided into sand flat, mud flat and mixed flat microfacies. The main gas reservoirs in the LX block are concentrated in the upper Palaeozoic, namely, the Permian, and are divided into six groups, which constitute the main research objects of this paper [33].

## 3. Materials and experiments

### 3.1. Materials

In this study, 18 complete cores with typical and representative diagenesis characteristics were collected from five wells in the tight sandstone in the Permian section of the upper Palaeozoic in the LX block [30]. The data were obtained from scanning electron microscopy (SEM), core casting thin-section analysis, conventional core physical property experimentation, core mercury intrusion experimentation, organic geochemical analyses and rock electrical conductivity experimentation. For the rock electrical conductivity experiments, a PLS-200 rock electrical conductivity meter was adopted. A small plunger with a radius of 3.5 cm was used to ensure the accuracy of each experiment. These experiments were carried out with a NaCl solution with a salinity of $20 \, \text{g l}^{-1}$ under a confining pressure of 5 MPa at room temperature. The core casting thin sections were obtained in accordance with the SY/T 5368–2000 Chinese national standard [34] under polarized light at room temperature. A JSM-5500LV scanning electron microscope was used for the microscopic observations and acquisition of representative photographs, and experimental measurements were carried out in an environment with a humidity of 40% and temperature of 25°C.

### 3.2. Experiments

To determine the mineral and rock compositions of the 18 rock samples, representative and fresh sections of the samples were selected to prepare core casting thin sections and SEM specimens. Polarization microscopy and SEM were used for microscopic observations. The core casting thin sections were examined in accordance with the SY/T 5368-2000 Chinese national standard [34]. To accurately study the diagenesis and pore structure, the polished thin sections were impregnated with coloured epoxy

resin for observation. The SEM was conducted in accordance with the SY/T 5162-1997 Chinese national standard. Panoramic and local images were obtained at different magnifications in conjunction with sample analyses and thin-section identification. Alizarin red staining was used to diagnose carbonate cement. The rock electrical parameters of the 18 rock samples were obtained in accordance with the SY/T 5385-2007 Chinese national standard, and the environmental temperature and humidity during the experiment were 25°C and 35–50% relative humidity, respectively. Eighteen plunger samples with lengths of approximately 35 mm and diameters of approximately 25 mm were drilled perpendicular to the drilling core column. The rock samples were saturated with a NaCl solution, and the resistivities corresponding to different saturation amounts were measured to obtain the rock electrical parameters. Finally, the capillary pressure curves of the rock samples were obtained by mercury injection. Referring to the SY/T 5346-2005 Chinese national standard ('Rock capillary pressure measurement'), the rock samples were cleaned and dried to constant weight, and their geometric dimensions, porosity and air permeability were measured. Then, the core chamber valve was opened, the mercury supplement valve was opened, and the height of the mercury cup was adjusted. The distance between the liquid level of the mercury cup and the suction valve corresponded to the height of the mercury column (approx. 760 mm) under the current atmospheric pressure; then, the isolation valve was opened, and the height of the mercury cup was adjusted again. At this time, the output value of the differential pressure sensor was between 28.00 and 35.00 cm; the suction valve was closed, the vacuum pump was closed and then opened again, and the mercury supplement valve was closed. The inlet valve of the high-pressure metering pump was closed, and the metering pump was adjusted so that the minimum gauge pressure was zero. According to the set pressure, the pressure and height of the mercury column in the mercury volume measuring tube after stabilization were recorded step by step until the highest set pressure was reached; the same procedure was conducted in reverse until the lowest set pressure was reached. Finally, according to the experimental results, the experimental data were sorted.

# 4. Results

## 4.1. Tight sandstones: texture and composition

The overall analysis based on the core casting thin section and SEM observations and statistics for the specimens from the study area shows that the main rock component is quartz, followed by rock fragments (figure 1). The rock fragments are abundant and have complex compositions; they are composed mainly of volcanic lithic fragments but include some metamorphic lithic fragments and sedimentary lithic fragments. The main lithologies in the LX block are lithic sandstone and sublitharenite with minor quartz arenite. According to the interstitial and rock structures, the study area has experienced strong diagenesis [35]; the compositions of the interstitial materials are diverse and complex, the reservoir is dense, and the particle size distribution is wide and non-uniform. The main pore types are dissolution intergranular pores, intragranular dissolution pores and residual intergranular pores, mainly secondary pores. Among them, intergranular pores, intragranular dissolved pores, intercrystalline pores and fractured pores are all developed.

In addition to identifying the mineral components of the rock samples, the diagenetic facies of 18 rock samples were determined by observing the main diagenesis and pore characteristics of each rock sample under a microscope.

## 4.2. Diagenetic minerals

### 4.2.1. Quartz

Quartz cements, in the form of quartz overgrowths (figure 2a) and authigenic quartz filling pores, are occasionally observed in thin section and SEM and are generally less than 4.0% of the whole rock in the study block. Quartz cements are often associated with various secondary dissolved pores and tend to occur in the Shiqianfeng Formation. This quartz cement formed after the pore-lining chlorite formed, implying that the pore-lining chlorite may have inhibited the growth of the quartz cement.

### 4.2.2. Carbonates

The main carbonate cement in the LX block is composed of calcite, ferric calcite and some ferric dolomite; according to the staining characteristics observed from core casting thin sections, it comprises mainly

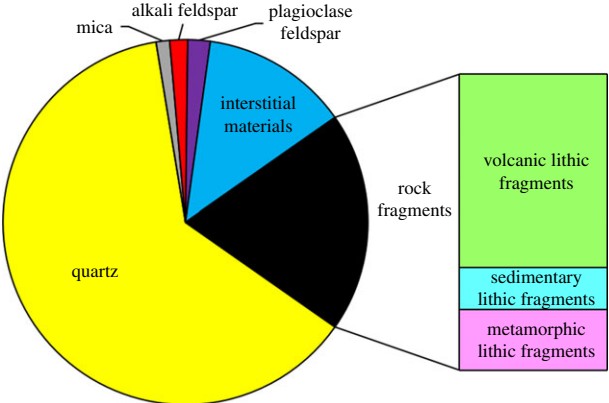

**Figure 1.** Histogram of the relative contents of the detrital composition.

early siderite, late calcite and minor iron-bearing dolomite (figure 2b,c). The early-stage cements consist of micritic calcite cement, usually without the metasomatism of clastic particles, and generally present a base-type or intergranular distribution. Such cementation plays an important role in enhancing the compaction resistance of sandy sediments, thereby preserving the intergranular volume and generating secondary pores for dissolution by acidic fluids in the later stage. Late-stage cements appear as fillers within intergranular or dissolution pores, which are mainly present in mineral components such as quartz and feldspathic and rock fragments. The composition of these late cements includes mainly ferrocalcite and minor ankerite, resulting in close contact between the particles and cement, which is usually composed of late-stage carbonate. This contact most commonly has a detrimental effect on reservoir properties.

### 4.2.3. Clay minerals

The main clay minerals in the studied rock include chlorite, illite and kaolinite, which have different textures; these minerals account for an average of 8.1% of the bulk samples (figure 2d–f). In the early stage of diagenesis, montmorillonite is converted into chlorite under iron-rich and manganese-rich conditions. Thus, the content of montmorillonite is controlled mainly by the precipitation of authigenic chlorite. The observed chlorite exists mainly in the form of granular films (pore linings) (figure 2d). Kaolinite is widely distributed throughout the LX block (figure 2e) and exists mainly in intergranular or heterogeneous groups, residual intergranular pores or feldspathic debris dissolution pores; kaolinite mostly fills granular dissolution pores. The formation of kaolinite is caused mainly by the dissolution of feldspar and dark minerals, which have the basic properties of volcanic lithic fragments, during early diagenesis (equations (4.1)–(4.5)). Plagioclase provides the necessary aluminium ions for the development of kaolinite. According to the SEM observations, the kaolinite within the intergranular pores is scaly (figure 2f). Illite forms mainly in pores exhibiting a fibrous appearance but has a relatively low abundance. The mixture of illite and montmorillonite and the mixture of illite and chlorite can be observed in the rock from the study area.

$$CaAlSi_2O_8(\text{calcium feldspar}) + 2H^+ + H_2O \rightarrow Al_2Si_2O_5(OH)_4(\text{kaolinite}) + Ca^{2+}, \quad (4.1)$$

$$2KAlSi_3O_8(\text{potassium feldspar}) + 2H^+ + H_2O \rightarrow Al_2Si_2O_5(OH)_4(\text{kaolinite}) + 4SiO_2$$
$$+ 4SiO_2(\text{siliceous}) + 2K^+, \quad (4.2)$$

$$2NaAlSi_3O_8(\text{albite}) + 2H^+ + H_2O \rightarrow Al_2Si_2O_5(OH)_4(\text{kaolinite}) + 4SiO_2(\text{siliceous}) + 2Na^+, \quad (4.3)$$

$$NaCa_2Fe_4Al_3Si_6O_{22}(OH)_2 + Ca_2MgSi_8O_{22}(OH)_2 + 26H^+ \rightarrow$$
$$NaAlSi_3O_8(\text{albite}) + Al_2Si_2O_5(OH)_4(\text{kaolinite}) + 9SiO_2 + 13H_2O$$
$$+ 5Mg^{2+}4Fe^{2+} + 4Ca^{2+} \quad (4.4)$$

$$\text{and} \qquad CaFe(SiO_3)_2 + CaMg(SiO_3)_2 + CaAl_2SiO_6 + 10H^+ \rightarrow$$
$$NaAlSi_3O_8(\text{albite}) + Al_2Si_2O_5(OH)_4(\text{kaolinite})$$
$$+ 3SiO_2(\text{kaolinite}) + 3SiO_2(\text{siliceous}) + 3H_2O + Mg^{2+}Fe^{2+} + 3Ca^{2+}. \quad (4.5)$$

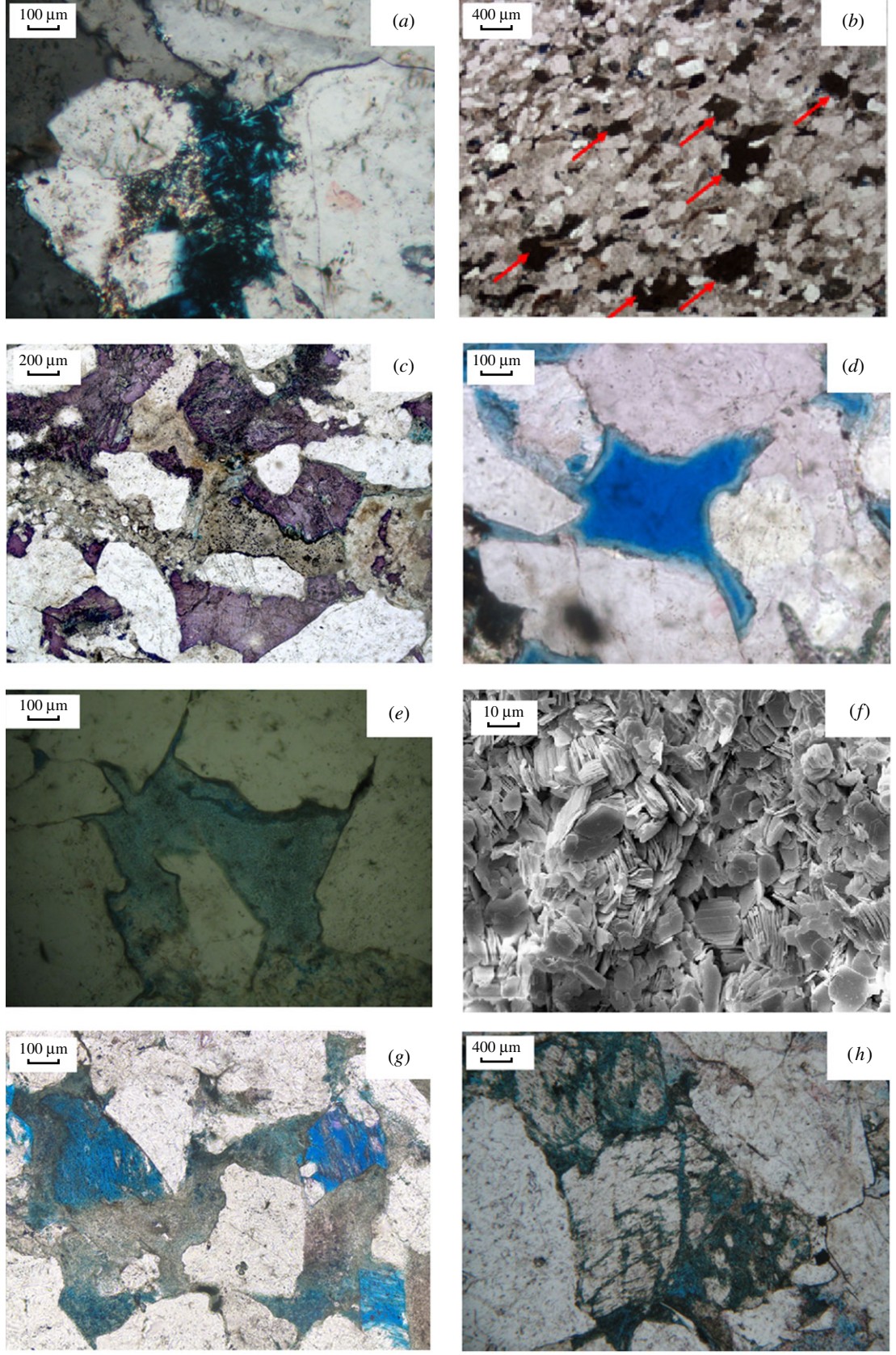

**Figure 2.** The characteristics of photographs in LX block. (*a*) The secondary growth of quartz, 100 µm. (*b*) Early siderite cementation, 400 µm. (*c*) Iron calcite cementation, 200 µm. (*d*) The chlorite film, 100 µm. (*e*) Kaolinite-filled intergranular pores, 100 µm. (*f*) Scaly kaolinite filling intergranular pores, 10 µm. (*g*) Feldspar dissolved, 100 µm. (*h*) The feldspar is dissolved to produce grid, sieve and palisade secondary pores, 400 µm.

## 4.3. Dissolution

The dissolution of soluble clastic particles is widely observed in the sandstone in the study area and is the reason for the formation of secondary pores. The main dissolution materials are feldspar skeleton particles and feldspar volcanic debris. Chemically unstable feldspars are susceptible to acidic fluids such as atmospheric precipitation and organic acids; potassium feldspar and albite react with $H_2O$ and carbon dioxide or react with organic acids (see equations (4.6)–(4.10)). Feldspar grains and oversized pores (mouldic pores) are commonly detected in thin-section analysis (figure 2$g$,$h$).

$$2NaAlSi_3O_8(albite) + 2CO_2 + 3H_2O \rightarrow Al_2Si_2O_5(OH)_4(kaolinite) + 4SiO_2 + 2Na^+ + 2HCO_3^-, \quad (4.6)$$

$$2KAlSi_3O_8(potassium\ feldspar) + 2H^+ + H_2O \rightarrow Al_2Si_2O_5(OH)_4(kaolinite) + 4SiO_2 + 2Na^+ + 2K^+, \quad (4.7)$$

$$3KAlSi_3O_8(potassium\ feldspar) + 2H^+ + H_2O \rightarrow KAl_3Si_3O_{10}(OH)_2(illite) + 6SiO_2 + H_2O + 2K^+ \quad (4.8)$$

$$2NaAlSi_3O_8(albite) + K^+ + 2H^+ + H_2O \rightarrow KAl_3Si_3O_{10}(OH)_2(illite) + 6SiO_2 + 3Na^+ + H_2O \quad (4.9)$$

$$\text{and} \qquad 3Al_2Si_2O_5(OH)_4(kaolinite) + 2K^+ \rightarrow 2KAl_3Si_3O_{10}(OH)_2(illite) + 3H_2O + 2H^+. \quad (4.10)$$

## 4.4. Classification of diagenetic facies

Diagenetic facies are named according to whether they increase or decrease reservoir capacity from a practical exploration perspective [36,37]. The characteristics of primary sedimentation and diagenesis were dominated by dissolution, cementation and changing porosity, and three diagenetic facies were identified: construction facies, cementation facies and destruction facies [30]. The construction facies includes sandstones with dissolved unstable components and sandstones with precipitated authigenic chlorite. Cementation facies includes sandstones with carbonate cements and sandstones with quartz cements, mainly sandstones with carbonate cements. Destruction facies includes sandstones with clay mineral cements. In the Discussion section, we analyse the pore characteristics and electrical conductivity characteristics of different diagenetic facies.

## 4.5. Tight sandstones: electrical characteristics

The straight lines in figure 3 indicate the relationship between the formation factor and porosity ranging from the ratio factors $a = 1$ and $m = 1$ to the ratio factors $a = 1$ and $m = 2.6$. The relationships between the formation factor and porosity show that in the case of $a = 1$, $m$ varies greatly from 1.3 to 2.6. As shown in figure 3, with decreasing porosity, the formation factors corresponding to low porosity deviate from the straight line corresponding to high porosity. Within the two sections with high and low porosities, the relationship between the formation factor and porosity exhibits non-Archie characteristics. According to the cross plot (figure 4) between the resistivity ($Rt$) and the water saturation ($Sw$) of the 18 rock samples, different rock samples have different relationships between $Rt$ and $Sw$, and non-Archie characteristics are also observed in the relationship between $Rt$ and $Sw$.

# 5. Discussion and derivation of the conductivity model

## 5.1. Analysis of factors influencing the electrical conductivity and pore characteristics of the diagenetic facies

The factors influencing the electrical conductivity of the diagenetic facies were comprehensively analysed based on core casting thin section and SEM observations and mercury injection and rock electrical conductivity experiments performed on the 18 rock samples. According to the rock electrical experiments, as mentioned above, the relationship between the formation factor and porosity in the study area does not conform to Archie's formula. Consequently, the individual factors affecting the conductivity were analysed from the perspectives of the lithological composition, pore structure and pore type. Among them, the component maturity and clay content are the two parameters representing the lithological composition, and the average pore throat radius obtained by mercury injection is chosen to characterize the pore structure (figure 5$a$–$c$). On the whole, the cementation exponent ($m$) is affected by the component maturity, clay content and average pore throat radius, but a strong correlation is not observed. To study the influence of the pore type on the rock electrical parameters, the dual-pore medium model proposed by Rasmus [38] was adopted to understand the

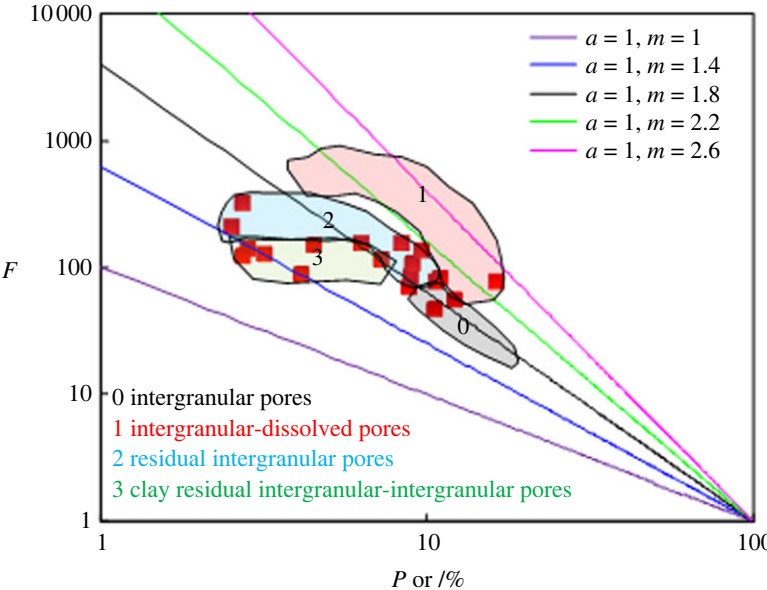

**Figure 3.** Relationship between *F* and Por.

controlling effect of the pore type on the electrical properties of rocks in tight gas reservoirs. According to core casting thin-section data, the pore structure of a tight gas reservoir can be generalized into intergranular pores, intergranular dissolved pores, residual intergranular pores and clay residual intergranular–intergranular pores. The relationships between the formation factors of different pore types and porosity (figure 3) can be simulated by the Rasmus model. Several pore types in figure 3 correspond to regions 0–3. Figure 5 further shows that the existence of dissolved pores increases the rock electrical parameter $m$; the smaller the matrix porosity is and the larger the dissolved pores are, the greater the increment in the rock electrical parameter $m$. Compared with rocks that have intergranular pores, rocks with the same porosity but clay residual intergranular–intergranular pores have a lower resistivity; that is, the rock electrical parameter $m$ decreases.

According to previous studies conducted in the Ordos Basin [32], diagenetic facies is a reasonable classification unit, and the difference in the pore system is one of the factors controlling the conductivity characteristics. Clay is another factor that influences the conductivity characteristics. Therefore, lithoelectrical parameters of the 18 rock samples from different diagenetic facies were analysed; the relationships between $m$ and $n$ for the different diagenetic facies are shown in figure 6. Overall, the $m$ value of the construction facies shown in figure 6 is higher than that of other facies. On the one hand, the average $m$ value of the construction facies conforms to the relationship between $m$ and the average pore radius $r_{mean}$ illustrated in figure 9. Because the average pore radius in the construction facies is higher than that of other facies, the value of parameter $m$ is also higher. On the other hand, the development of dissolution pores in the construction facies and the increase in dissolution pores both increase the $m$ value.

It is necessary to clarify the differences in electrical conductivity in terms of the lithoelectrical parameters of diagenetic facies because the conductivity characteristics of the different diagenetic facies vary. To clarify the electrical conductivity characteristics of the three diagenetic facies, first, the factors that influence the electrical conductivity of each diagenetic facies were analysed. According to a similar study in the Ordos Basin [32], within each diagenetic facies, the lithoelectrical parameters are affected by a single main factor, and $m$ in the LX block is controlled mainly by the pore structure: differences in the pore structures among diagenetic facies cause the parameter $m$ to vary. To better study the conductive characteristics of diagenetic facies, we first systematically performed research on the pore characteristics of those diagenetic facies and then established conductivity models of the diagenetic facies.

Porosity can reflect the reservoir capacity of a rock, while the permeability, pore throat size, connectivity and particle size distribution determine the flow characteristics. Therefore, the pore size, pore structure, pore type, interconnectivity and relationships between these factors of various diagenetic facies must be studied to establish pore structure models of those facies. First, the distributions of the porosity and permeability of the tight sandstone cores are shown in figure 7. The

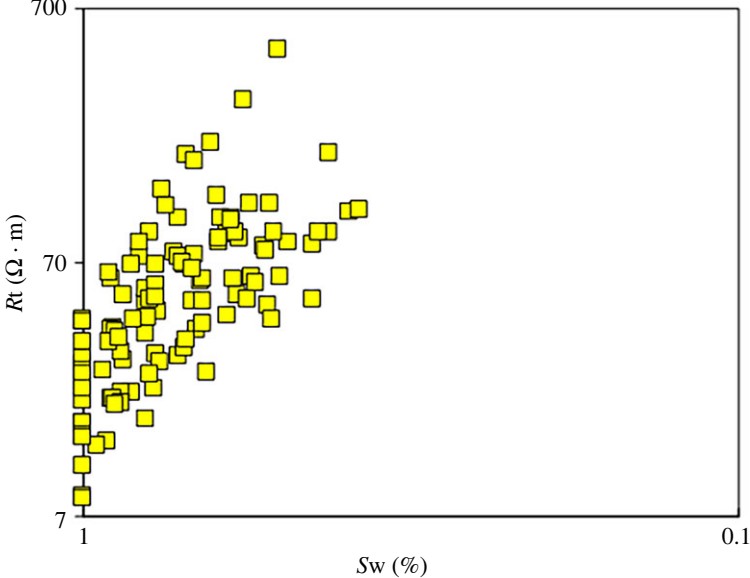

**Figure 4.** Relationship between *R*t and *S*w.

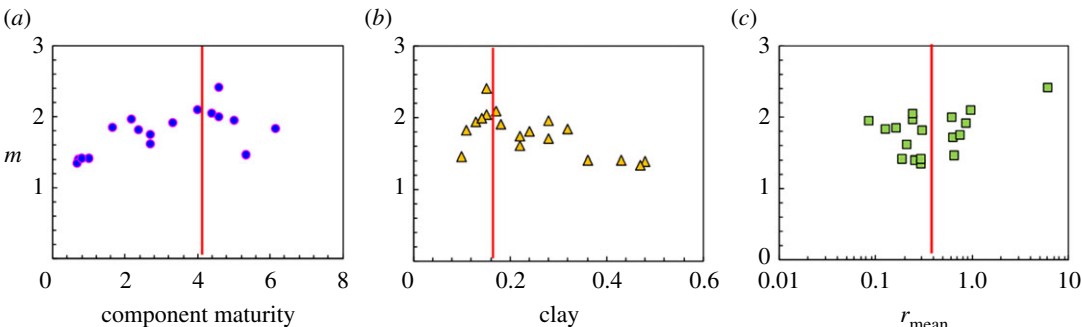

**Figure 5.** Relationship between cementation index and component maturity, clay and $r_{mean}$.

porosity of the cementation facies varies between 2% and 10%, although it is generally concentrated at 8%, and the permeability ranges between $0.2 \times 10^{-3}$ and $1 \times 10^{-3}\ \mu m^2$. The porosity of the destruction facies varies between 2% and 9% with an average of 4%, and the permeability varies between $0.01 \times 10^{-3}$ and $0.2 \times 10^{-3}\ \mu m^2$, with an average of approximately $0.1 \times 10^{-3}\ \mu m^2$. The porosity of the construction facies ranges between 5% and 16%, with most of the porosity values exceeding 7%, and the permeability varies between $0.1 \times 10^{-3}$ and $10 \times 10^{-3}\ \mu m^2$, with most of the permeability values exceeding $1 \times 10^{-3}\ \mu m^2$. The physical properties of the construction facies are superior to those of the other facies.

Next, the parameters that reflect the pore structures and pore types of various diagenetic facies were calculated (figures 8–11) according to the mercury injection and core casting thin-section data. Figure 8 reveals that the degrees of mercury distortion of the three diagenetic facies are all distributed between −1 and 1. The values of $r_{35}$ and $r_{mean}$ for the cementation facies are smaller than those for the construction facies, and the values of *P*d and *S*p are low (figure 9). Most of the values exhibit concentrated distributions. These mercury injection parameters indicate that the physical properties of the cementation facies are somewhat favourable. By contrast, among the results of all the diagenetic facies, the values of $r_{35}$ and $r_{mean}$ for the destruction facies are relatively small, while the corresponding *P*d value is relatively low. For the construction facies, the degree of mercury distortion (skp) is greater than 0, which is coarse. Additionally, the values of $r_{35}$ and $r_{mean}$ are relatively large, and that of *P*d is relatively low (figures 9 and 10), while the value of *S*p is comparatively high. The quartz content is higher in the construction facies than in the other two diagenetic facies, and the content of rock fragments is relatively high in the destruction facies (figure 11).

Overall, the mercury injection parameters indicate that the pore structures of the construction facies are more favourable than those of the other two diagenetic facies and that its pore distribution is

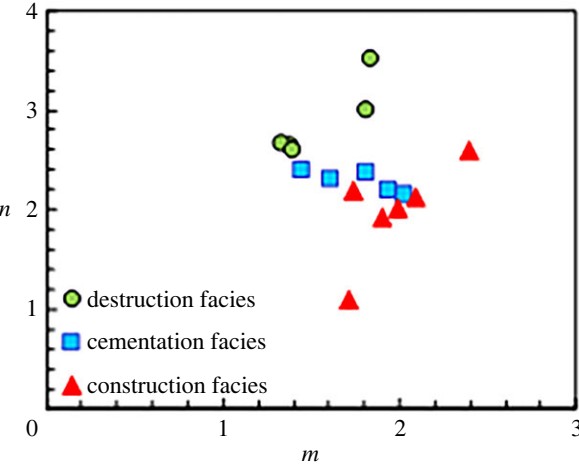

**Figure 6.** Relationship between $m$ and $n$ in three types of diagenetic facies.

relatively non-uniform. Moreover, the pore space of the construction facies is composed of residual intergranular pores and dissolution pores, and the overall pore structure is relatively complex. By contrast, for the cementation facies, while the pore distribution is asymmetric, the pore space is dominated by residual intergranular pores, and there is mostly a single pore type. Additionally, the pore throat distribution is relatively concentrated, and the overall pore structure is relatively simple. For the destruction facies, the pore distribution is relatively asymmetrical, and its physical properties are poor; the pore space is composed mainly of residual intergranular pores and small dissolved pores, and the pore types are relatively complicated.

According to the pore characteristics of the diagenetic facies in the LX block, schematic maps of the diagenetic facies were drawn. The schematic map of the construction facies shown in figure 12*a* was established according to the core casting thin section, mercury injection and other experimental analysis results. Dissolution was the main diagenetic process in this facies. This facies also includes chlorite as the main cement, which exists mainly as pore linings, greatly improving the compaction resistance of the reservoir and thus the preservation of primary pores. Strong dissolution led to the formation of many secondary pores and increased both the reservoir space and the reservoir interpore connectivity. The porosity is above 7%, and most permeability values exceed $1 \times 10^{-3}$ µm$^2$ (figure 7). Overall, construction facies with relatively high porosity and permeability properties, also called sweet spots, are highly sought after by petroleum geologists. By contrast, the cementation facies (figure 12*b*) is composed mainly of carbonate cement and is developed mostly in thick sandstone due to fluid migration. Because the fine sandstones experienced early compaction, the physical properties and pore structures of the fine sandstones are poor, as pore fluids cannot migrate effectively within the pores. Therefore, the quartz cement and carbonate cement contents are lower in the fine sandstones than in the coarse-grained sandstones. Finally, a schematic map of the destruction facies is shown in figure 12*c*. The main characteristics of the destruction facies are sedimentation and filling with kaolinite and sericite. The development of a large amount of kaolinite often signifies a reduction in primary porosity and the generation of a large number of secondary dissolution pores. The reaction that precipitates sericite produces small sediments that destroy the porosity and pore throats. Hence, the main processes responsible for forming the destruction facies are the autogenetic cementation of kaolinite and the sericitization of volcanic debris in pores.

## 5.2. Conductivity characteristics

For reservoirs with very complex pore systems, the application of Archie's formula is limited. This restriction is reflected mainly in the significant differences among the pore index magnitudes of various types of reservoirs. In other words, the complexity of the pore structure intensifies the variation in the cementation exponent $m$ in the classic Archie formula, which affects the accuracy of the saturation calculation. According to the analysis and summary of the physical properties and the pore structure characteristics of the various diagenetic facies within the LX block, these three diagenetic facies clearly exhibit different pore structure characteristics [39]. Therefore, according to the characteristics of the pore structures of the three diagenetic facies, conductivity characteristics are

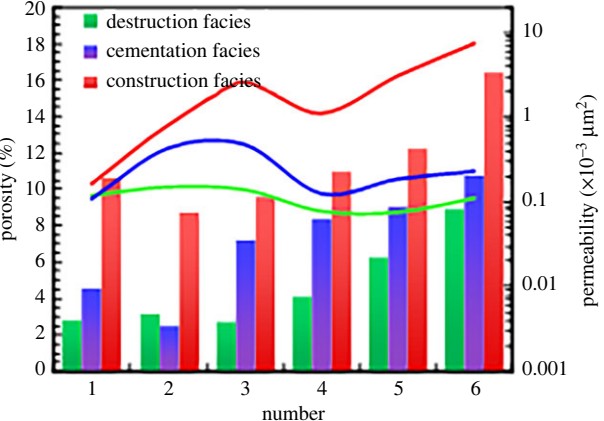

**Figure 7.** Distribution diagram of porosity and permeability of three types of diagenetic facies.

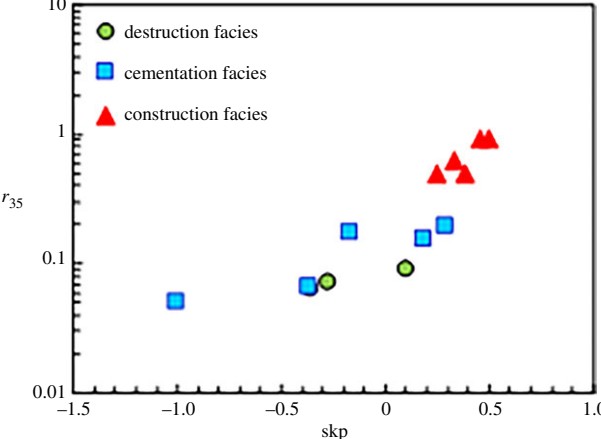

**Figure 8.** Cross plot of diagenetic facies by the mercury distortion (skp) and $r_{35}$.

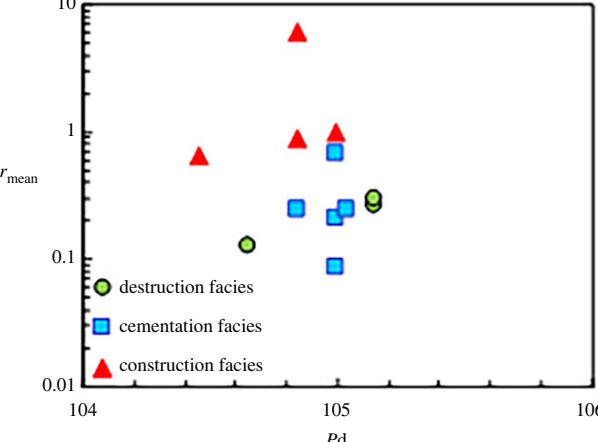

**Figure 9.** Cross plot by displacement pressure (*P*d) and $r_{mean}$ of diagenetic facies.

discussed, and conductivity models suitable for these diagenetic facies are deduced. The nomenclature is shown in table 1.

### 5.2.1. Destruction facies

According to core casting thin-section observations and mercury injection experiments, the destruction of clay residual intergranular pores and intergranular pores [40] can be considered in a conductivity model

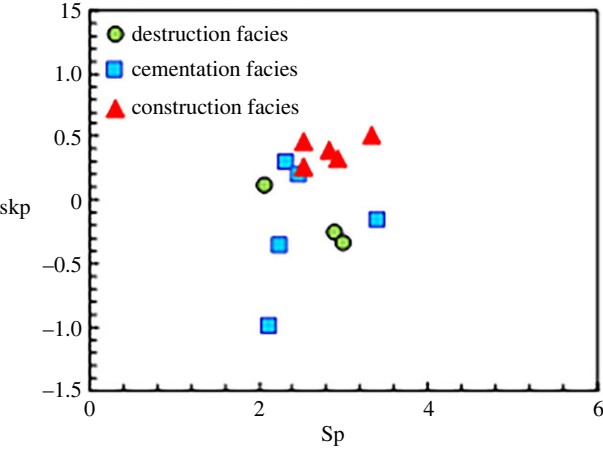

**Figure 10.** Cross plot by Sp and the mercury distortion (skp) of three types diagenetic facies in LX block.

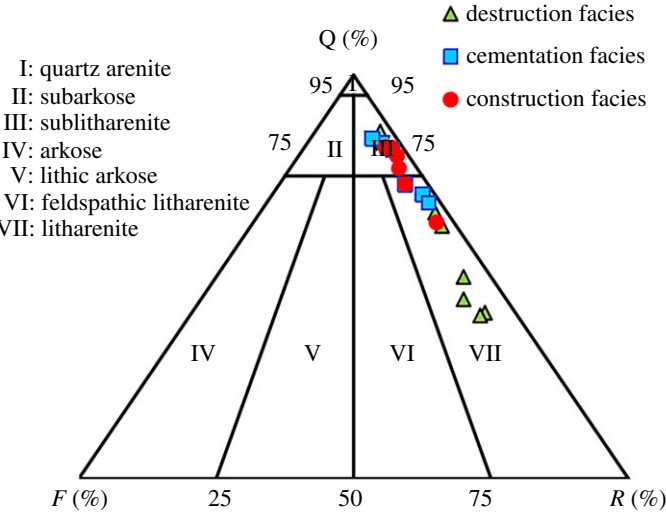

**Figure 11.** Triangle lithology map in LX block.

of this dual-pore system. This model, which can be adopted for matrix pores, is an interconnecting system of porosity composed of intergranular pores (clay residual intergranular pores); furthermore, this model can be used to establish conductivity models [41]. These conductivity models assume that the destruction facies of subsurface rocks is completely saturated with water and that the rock resistivity $R_{\text{des}}$ is calculated as follows:

$$R_{\text{des}} = \nu_{\text{nc}} \Phi R_w + (1 - \nu_{\text{nc}} \Phi) R_o. \tag{5.1}$$

When the pore type in the rock consists of only matrix pores,

$$F = \Phi_b^{-m_b}. \tag{5.2}$$

When the pore type within the rock consists of only composite pores,

$$R_{\text{des}} = F_t R_w \tag{5.3}$$

and

$$F_t = \Phi^{-m}. \tag{5.4}$$

Then, equation (5.3) can be substituted into equation (5.1),

$$F_t R_w = \nu_{\text{nc}} \Phi R_w + (1 - \nu_{\text{nc}} \Phi) R_o. \tag{5.5}$$

**Table 1.** Nomenclature.

| | |
|---|---|
| $A_b$ | area of matrix |
| $A_{con}$ | area of volume |
| $A_d$ | area of interconnected pores |
| $F$ | formation factor |
| $F_t$ | formation factor of composite system |
| $L_b$ | length of matrix |
| $L_{con}$ | length of volume |
| $L_d$ | length of interconnected pores |
| $m_b$ | matrix cementation exponent |
| $m$ | cementation exponent of destruction facies |
| $M$ | cementation exponent of construction facies |
| $R_{des}$ | resistivity of the composite system (matrix pores and unconnected pores (clay intergranular pores)) at 100% saturated water |
| $r_{con}$ | resistance of composite system (matrix pores and interconnected pores (dissolved pores)) |
| $r_b$ | resistance of matrix |
| $r_d$ | resistance of interconnected pores |
| $R_{con}$ | resistivity of the composite system (matrix pores and interconnected pores (dissolved pores)) |
| $R_b$ | resistivity of matrix |
| $R_d$ | resistivity of interconnected pores |
| $R_w$ | formation water resistivity at formation temperature |
| $R_o$ | resistivity at reservoir temperature when the 100% saturated resistivity is $R_w$, the resistivity of the matrix system |
| $v_{nc}$ | ratio of non-connected pores |
| $v_b$ | ratio of matrix pores |
| $v_d$ | ratio of interconnected pores |
| $\phi$ | porosity |
| $\phi_b$ | matrix porosity related to the total volume of the matrix system |

Dividing both sides of equation (5.5) by $R_w$, results in the following expression

$$F_t = v_{nc}\Phi R_w + (1 - v_{nc}\Phi)R_o/R_w$$
$$= v_{nc}\Phi + (1 - v_{nc}\Phi)F. \tag{5.6}$$

Then, equation (5.2) and equation (5.4) can be substituted into equation (5.6),

$$\Phi^{-m} = v_{nc}\Phi + (1 - v_{nc}\Phi)\Phi_b^{-m_b}. \tag{5.7}$$

Next, the cementation exponent $m$ of the destruction facies can be expressed as

$$m = \frac{\log[v_{nc}\Phi + (1 - v_{nc}\Phi)\Phi_b^{-m_b}]}{-\log \Phi}. \tag{5.8}$$

Lucia's vug porosity ratio can be expressed as

$$v_{nc} = \frac{\Phi_{nc}}{\Phi} = \frac{\Phi - \Phi_m}{\Phi} = \frac{\Phi - \Phi_b}{\Phi(1 - \Phi_b)}. \tag{5.9}$$

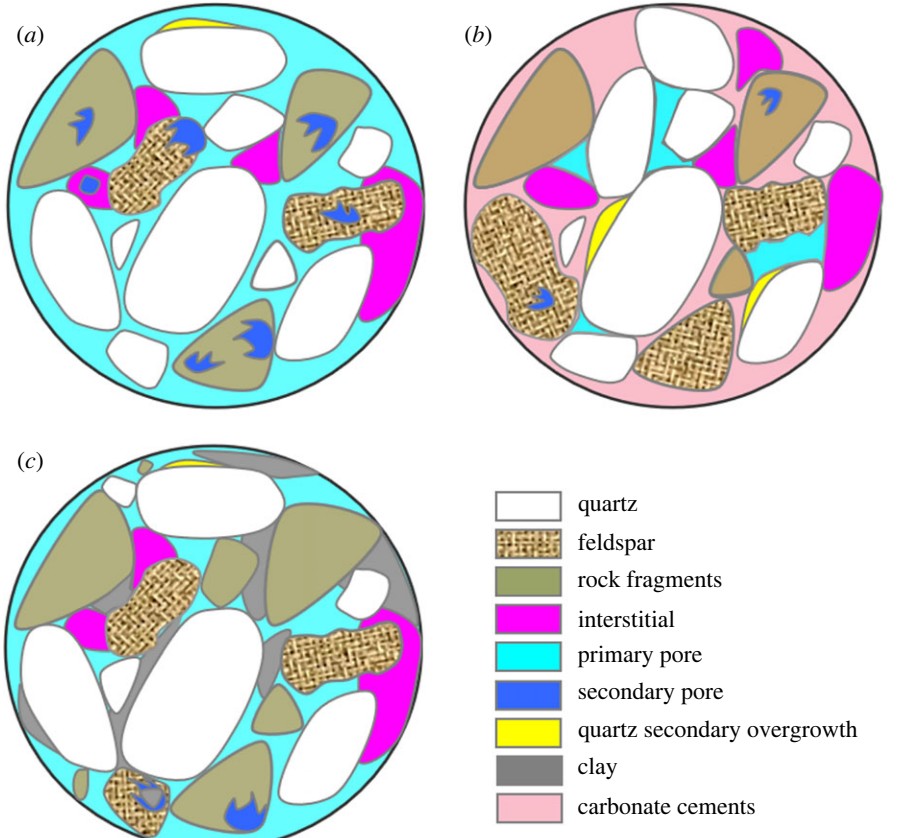

**Figure 12.** Schematic by diagenetic facies. (*a*) Construction facies. (*b*) Cementation facies. (*c*) Destruction facies.

### 5.2.2. Construction facies

The overall reservoir conditions of the construction facies are superior to those of the destruction facies and the cementation facies. The pore throats are located mainly between dissolved feldspar pores and intergranular pores, and the construction facies can be simplified into a pore system composed of matrix pores and interconnected pores (dissolved pores). With the parallel conductivity model [38], the overall resistance $r_{\mathrm{con}}$ of the rock has the following relationship:

$$\frac{1}{r_{\mathrm{con}}} = \frac{1}{r_b} + \frac{1}{r_d}, \tag{5.10}$$

where

$$r = R\frac{L}{A}. \tag{5.11}$$

Then, equation (5.10) can be expressed as

$$\frac{A_{\mathrm{con}}}{R_{\mathrm{con}}L_{\mathrm{con}}} = \frac{A_b}{R_b L_b} + \frac{A_d}{R_d L_d}. \tag{5.12}$$

Multiplying both sides of equation (5.12) by $L_{\mathrm{con}}^2$ yields

$$\frac{A_{\mathrm{con}}L_{\mathrm{con}}}{R_{\mathrm{con}}} = \frac{A_b L_{\mathrm{con}}}{R_b}\frac{L_{\mathrm{con}}}{L_b} + \frac{A_d L_{\mathrm{con}}}{R_d}\frac{L_{\mathrm{con}}}{L_d}. \tag{5.13}$$

Here, we define the entire rock volume as 1 and $v_b = L_b/L_{\mathrm{con}}$; then, equation (5.13) becomes

$$\frac{1}{R_{\mathrm{con}}} = \frac{A_b L_{\mathrm{con}}}{R_b v_b} + \frac{A_d L_{\mathrm{con}}}{R_d v_d}, \tag{5.14}$$

where $R_b$ can be expressed as $R_w/\varphi_b^M S_w^{Nb}$. Thus, equation (5.14) can be expressed as

$$\frac{1}{R_{\mathrm{con}}} = \frac{V_b \varphi_b^{M_b} S_w^{Nb}}{R_w} + \frac{V_d S_w^{Nd}}{R_w v_d^2}, \tag{5.15}$$

$V_d$ can be expressed as the difference between the total porosity and primary porosity,

$$V_d = \phi - \phi_b \tag{5.16}$$

and

$$V_b = 1 - V_d = 1 - (\phi - \phi_b). \tag{5.17}$$

Then, equation (5.16) and equation (5.17) can be substituted into equation (5.15):

$$\frac{[1 - (\varphi - \phi_b)][\phi_b^{M_b} S_{wb}^{Nb}]}{R_w} + \frac{1}{R_{\mathrm{con}}} = \frac{(\varphi - \phi_b) S_{wf}^{Nf\,(1/v_d^2)}}{R_w}. \tag{5.18}$$

When the rock is completely saturated with water, $R_w$ can be expressed as $R_o$; then,

$$\frac{R_w}{R_o} = [1 - (\varphi - \phi_b)] \phi_b^{M_b} + (\varphi - \phi_b) \frac{1}{v_d^2}, \tag{5.19}$$

where

$$\frac{R_w}{R_o} = \frac{1}{F'} = \varphi^{M'}. \tag{5.20}$$

Next, equation (5.20) can be substituted into equation (5.19),

$$M = \frac{\log\{[1 - (\varphi - \phi_b)] \phi_b^{M_b} + [\varphi - \phi_b)(1/v_d^2)]\}}{\log \varphi}. \tag{5.21}$$

Letting $(1/v_d^2) = 1$,

$$M = \frac{\log\{[1 - (\varphi - \phi_b)] \phi_b^{M_b} + [(\varphi - \phi_b)]\}}{\log \varphi}. \tag{5.22}$$

### 5.2.3. Cementation facies

According to the previous analysis, the cementation facies is composed of primarily residual intergranular pores, and its pore structures are relatively simple. From a pore structure perspective, the cementation facies pore throat distribution is relatively concentrated. This analysis demonstrates that the cementation exponent of the cementation facies is controlled by (and has a close relationship with) the porosity (figure 13).

$$m = 0.3959 * \ln(\emptyset) + 1.0586. \tag{5.23}$$

Where $\emptyset$ is porosity.

## 5.3. Analysis of the rationality of the model and factors that influence the conductivity characteristics

To fully verify the reliability and rationality of the model, this section focuses on three parts: the rationality of the model, the influences of the pore types and structures on the conductivity characteristics of the rock, and an evaluation of the conductivity models for the diagenetic facies. To analyse the rationality of the model and the influences of the parameters in each model on the final cementation exponent, the relationship between the cementation exponent and various pores was schematically illustrated using a simulation method and an intersection diagram.

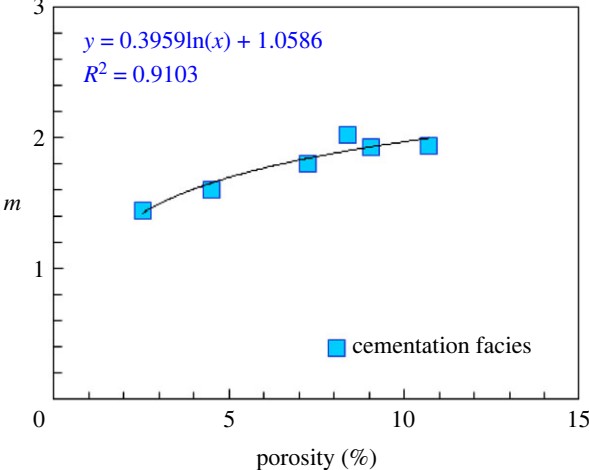

**Figure 13.** Relationship between cementation exponent and porosity.

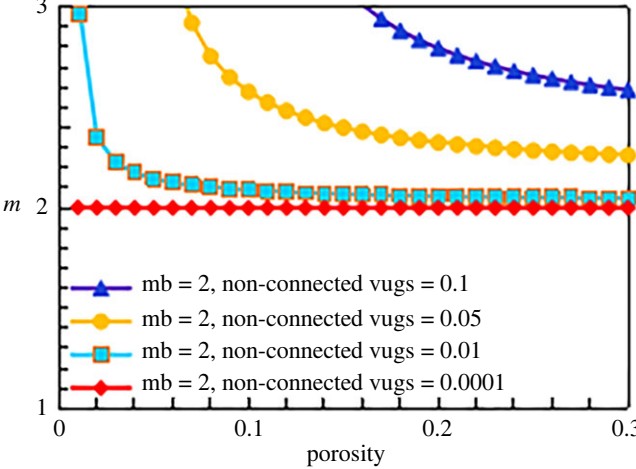

**Figure 14.** The relationship between cementation exponent $m$ and total porosity with different content of non-connected pores (mb = 2).

### 5.3.1. Destruction facies

According to the cementation exponent formula for the destruction facies (equation (5.9)), the $m$ value of the destruction facies is affected mainly by two pore types: non-connected pores and matrix pores. Based on the experimental data and experience, the conventional empirical value of 2.0 is adopted for the matrix cementation exponent $m_b$ (figure 14). In figure 14, the relationship between the cementation exponent and the total porosity is obtained when the non-connected porosity differs. The line of blue triangles indicates the destruction facies conductivity model when the non-connected porosity is 0.1, the line of yellow circles represents the destruction facies conductivity model when the non-connected porosity is 0.05, and the line of blue squares and the line of red diamonds are the destruction facies conductivity models when the non-connected porosities are 0.01 and 0.0001, respectively. When the non-connected porosity is small (line of red diamonds, non-connected porosity of 0.0001), the influence of the non-connected porosity on the cementation exponent is very small, and the model can be simplified into the case where only intergranular pores exist; the corresponding value of $m$ is approximately 2. As the non-connected porosity increases (from the line of blue squares to the line of blue triangles), the content of non-connected pores gradually affects the cementation exponent of the rock. With the same total porosity of the rock, a higher content of non-connected pores corresponds to a larger cementation exponent. This finding is consistent with observations from experimental data and the results of other scholars [6]. When the number of non-connected pores in the system is large, for example, 0.05 (line of yellow circles), the cementation exponent decreases as the total porosity

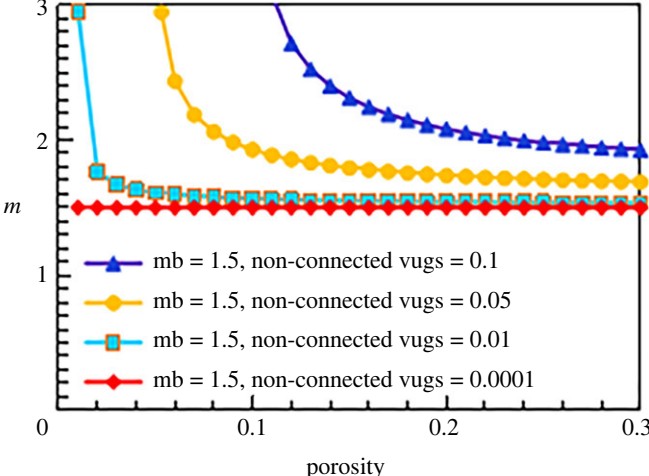

**Figure 15.** The relationship between cementation exponent $m$ and total porosity with different content of non-connected pores (mb = 1.5).

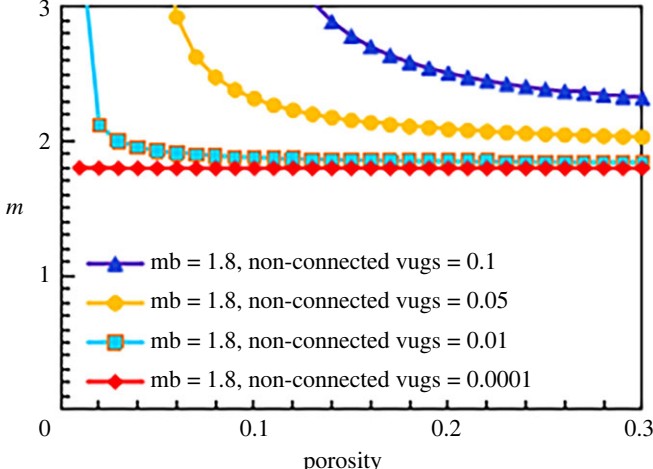

**Figure 16.** The relationship between cementation exponent $m$ and total porosity with different content of non-connected pores (mb = 1.8).

increases. When the total porosity is small, namely, when non-connected pores occupy a relatively high proportion of the rock, the cementation exponent decreases rapidly. With an increase in the total porosity, the proportion of non-connected pores decreases, and the change in the $m$ value becomes small.

When the matrix cementation exponents are 1.5, 1.8 and 2.5, the results are similar, as shown in figures 15–17.

### 5.3.2. Construction facies

According to the calculation formula for the cementation exponent of the construction facies (equation (5.23)), the cementation exponent $m$ of the construction facies is affected mainly by two types of pores: intergranular pores and dissolution feldspar pores. Taking a matrix cementation exponent of 2.0 as an example, the results are shown in figure 18, which displays the relationship between the cementation exponent and the porosity for different dissolution pore contents. The line of blue triangles is the conductivity model of the construction facies in the case with a dissolution porosity of 0.1; the line of yellow circles is the conductivity model of the construction facies when the dissolution porosity is 0.05; and the line of blue squares and the line of red diamonds are the conductivity models of the construction facies when the dissolution porosity values are 0.01 and 0.001, respectively. When the dissolution porosity is small (line of red diamonds, dissolution porosity = 0.001), the influence of the dissolution pores on the cementation exponent is very small, and the model can be simplified so

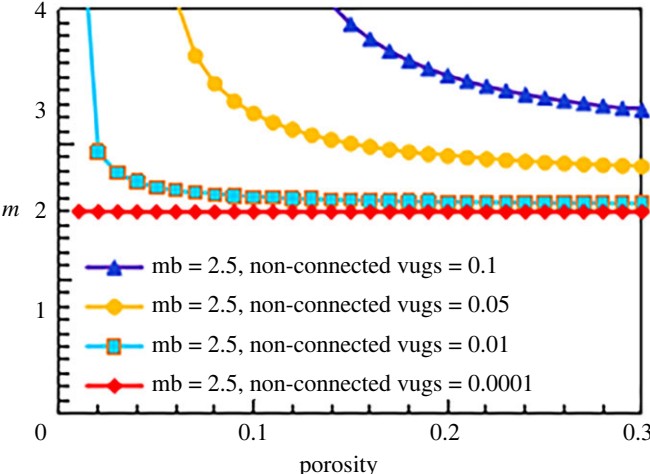

**Figure 17.** The relationship between cementation exponent $m$ and total porosity with different content of non-connected pores (mb = 2.5).

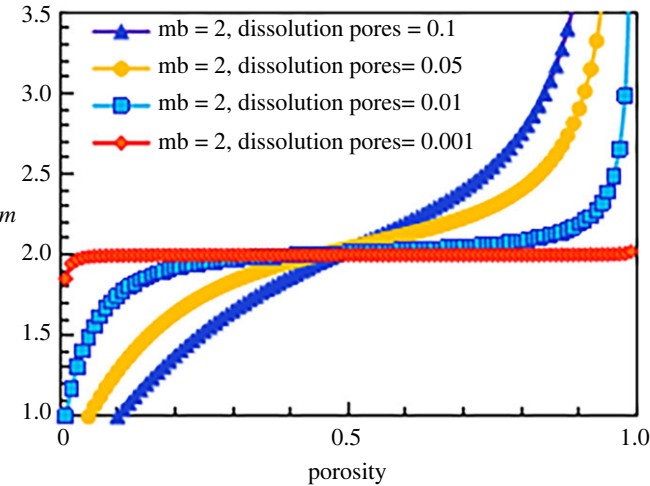

**Figure 18.** The relationship between cementation exponent $m$ and total porosity at different dissolution pore contents (mb = 2).

that only intergranular pores exist; thus, the value of $m$ is approximately 2. As the dissolution porosity increases (from the line of blue squares to the line of blue triangles), the content of the dissolution porosity gradually affects the cementation exponent of the rock. For a certain porosity value, as the proportion of dissolution porosity increases, the dissolution porosity increases and the cementation exponent decreases, which is consistent with the observed experimental results and other scholars' research results. However, when the porosity increases indefinitely (a purely theoretical situation), such as in the case of a certain total pore number, the dissolution porosity and the cementation exponent both increase.

According to this method, the results corresponding to cementation exponents of 1.8, 2.5 and 3 for the simulated construction facies are shown in figures 19–21, respectively.

## 5.4. Model validation

In this paper, data from 18 core samples acquired from tight sandstone reservoirs in the Ordos Basin were used to verify the electrical conductivity models of three diagenetic facies. The diagenetic facies types of the 18 core samples were determined based on microscopic core casting thin section and SEM observations, pore structure experiments such as mercury injection experiments, nuclear magnetic resonance analysis and overpressured permeability calculations. Six diagenetic rock samples pertain to each of the three facies identified (construction facies, cementation facies and destruction facies). The parameters of other conductivity models were also obtained by laboratory measurements. Among these samples, the porosity was obtained by physical experiments; the matrix porosity, dissolution

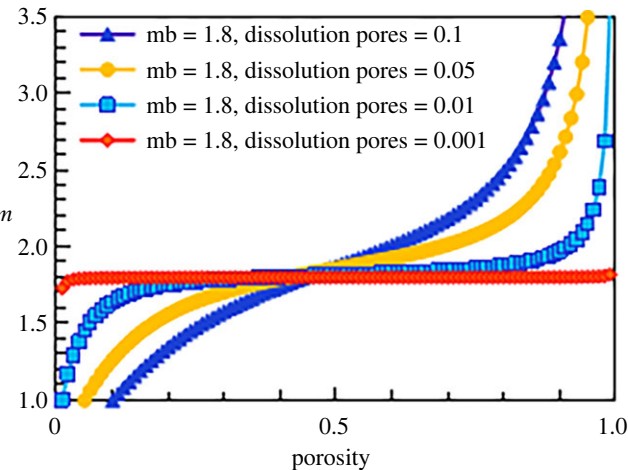

**Figure 19.** The relationship between cementation exponent *m* and total porosity at different dissolution pore contents (mb = 1.8).

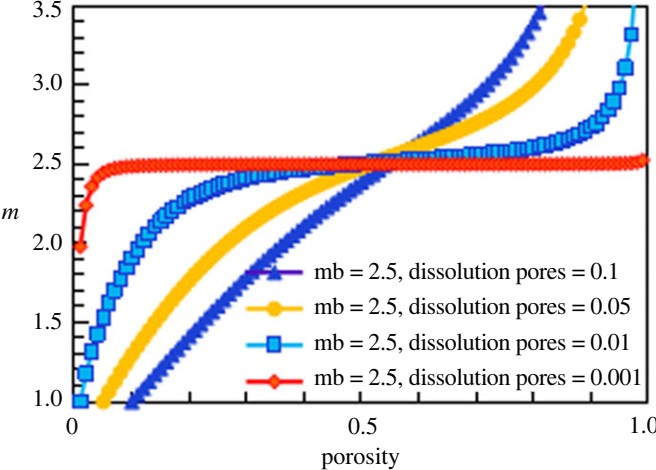

**Figure 20.** The relationship between cementation exponent *m* and total porosity at different dissolution pore contents (mb = 2.5).

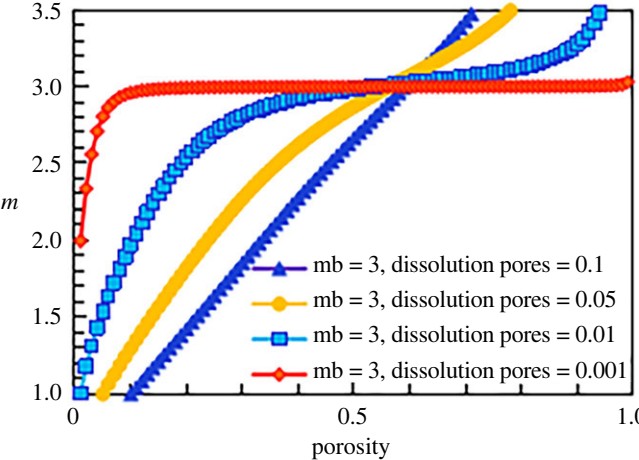

**Figure 21.** The relationship between cementation exponent *m* and total porosity at different dissolution pore contents (mb = 3).

porosity and non-connected porosity were obtained by core casting thin-section analysis; and the cementation exponent was obtained by electrical conductivity experiments. The specific experimental data are shown in table 2.

Regarding the corresponding conductivity models of the various diagenesis facies, the conductivity model of a dual-pore system was used for the destruction facies, and the construction facies was

**Table 2.** Physical properties of the tight samples.

| no. | depth (m) | diagenetic facies | quartz (%) | feldspar (%) | rock fragments (%) | porosity (%) | F | m (a = 1) | matrix pore (%) | non-connected pore (%) | dissolution pore (%) |
|---|---|---|---|---|---|---|---|---|---|---|---|
| 1 | 1519 | Destruction facies | 62.61 | 2.61 | 34.78 | 8.9 | 83.52 | 1.83 | 2.54 | 6.36 | / |
| 2 | 1707.62 | Destruction facies | 50.00 | 5.00 | 45.00 | 4.1 | 85.82 | 1.39 | 2.46 | 1.64 | / |
| 3 | 1794.98 | Destruction facies | 86.02 | 2.15 | 11.83 | 6.3 | 149.39 | 1.81 | 1.80 | 4.50 | / |
| 4 | 1956.39 | Destruction facies | 41.11 | 5.56 | 53.33 | 2.82 | 135.64 | 1.38 | 1.41 | 1.41 | / |
| 5 | 1956.99 | Destruction facies | 40.45 | 6.74 | 52.81 | 2.72 | 118.55 | 1.32 | 1.36 | 1.36 | / |
| 6 | 1955.39 | Destruction facies | 44.44 | 7.78 | 47.78 | 3.18 | 122.71 | 1.39 | 1.27 | 1.91 | / |
| 7 | 1854.88 | Cementation facies | 70.45 | 2.27 | 27.27 | 7.25 | 113.62 | 1.80 | / | / | / |
| 8 | 1647.38 | Cementation facies | 72.92 | 4.17 | 22.92 | 4.51 | 144.94 | 1.61 | / | / | / |
| 9 | 1217.24 | Cementation facies | 68.42 | 2.11 | 29.47 | 10.7 | 76.61 | 1.94 | / | / | / |
| 10 | 1791.67 | Cementation facies | 81.52 | 2.17 | 16.30 | 8.4 | 150.41 | 2.02 | / | / | / |
| 11 | 1646.42 | Cementation facies | 83.33 | 3.13 | 13.54 | 9.05 | 103.85 | 1.93 | / | / | / |
| 12 | 1470.8 | Cementation facies | 84.27 | 4.49 | 11.24 | 2.53 | 203.18 | 1.45 | / | / | / |
| 13 | 1550.83 | Construction facies | 80.00 | 2.11 | 17.89 | 9.6 | 133.40 | 2.09 | 5.76 | / | 3.84 |
| 14 | 1284.11 | Construction facies | 82.11 | 3.16 | 14.74 | 10.9 | 81.80 | 1.99 | 4.42 | / | 6.48 |
| 15 | 1552.69 | Construction facies | 82.11 | 2.11 | 15.79 | 16.4 | 75.55 | 2.39 | 10.58 | / | 5.82 |
| 16 | 1716.78 | Construction facies | 63.49 | 3.17 | 33.33 | 10.6 | 45.68 | 1.70 | 4.54 | / | 6.06 |
| 17 | 1285.51 | Construction facies | 72.92 | 4.17 | 22.92 | 8.76 | 68.02 | 1.73 | 4.17 | / | 4.59 |
| 18 | 1470.23 | Construction facies | 76.92 | 3.30 | 19.78 | 12.2 | 54.33 | 1.90 | 6.10 | / | 6.10 |

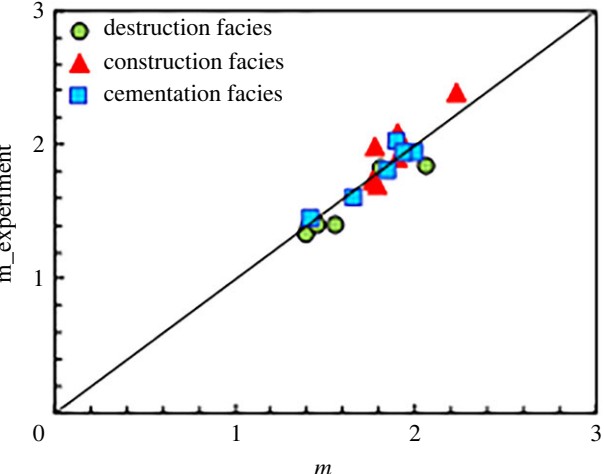

**Figure 22.** Comparison between evaluation results and experimental results of parameter *m*.

described with a pore system composed of matrix pores and interconnected pores (dissolved pores). The parallel conductivity model was used for the construction facies. The cementation facies with a relatively simple pore type was evaluated by the porosity-cementation exponent model. The cementation exponents were calculated with equations (5.8), (5.22) and (5.23), respectively. Then, the cementation exponents measured in the laboratory were compared, as shown in figure 22. According to the evaluation results of the three diagenetic facies, the cementation exponent of each diagenetic facies is in good agreement with the experimental results.

# 6. Conclusion

For complex tight sandstone reservoirs that have been subjected to strong diagenesis and exhibit complex electrical conductivity characteristics, diagenetic facies is a suitable classification unit. Diagenetic facies are named according to whether they increase or decrease reservoir capacity from a practical exploration perspective. The reservoir was divided into three types of diagenetic facies: construction facies, cementation facies and destruction facies. Accordingly, the main diagenetic properties, pore sizes, pore structures, pore types, interconnectivities and relationships among these factors of various diagenetic facies were also studied. Then, using the conductivity model of a dual-pore system, the destruction facies was simplified into a pore system consisting of matrix pores and non-connected pores (clay intercrystalline pores) that can be described using a series conductivity model. By contrast, the construction facies was simplified into a pore system consisting of matrix pores and interconnected pores (dissolved pores), as determined by a parallel conductivity model. The pore structures and pore types of the cementation facies in the study area are simple, and the cementation exponent is controlled by the amount of porosity within the rock. A simulation method was adopted to examine the relationships between various diagenetic facies conductivity models and different types of porosity by means of intersection diagrams. For the destruction facies, when the number of non-connected pores in the system is large, the cementation exponent decreases as the total porosity increases. When the total porosity is small, namely, when non-connected pores occupy a relatively high proportion of the rock, the cementation exponent decreases rapidly. With an increase in the total porosity, the proportion of non-connected pores decreases, and the change in the *m* value becomes small. For the construction facies, when the dissolution porosity is small, the influence of the dissolution pores on the cementation exponent is very small. As the dissolution porosity increases, as the proportion of dissolution porosity increases, the cementation exponent decreases. These diagenetic facies conductivity models are reasonable and can characterize the conductivity characteristics of the various diagenetic facies.

Data accessibility. Zhou, Xueqing (2021), The simulation results of relationship between cementation exponent *m* and total porosity of different diagenetic facies with different content of parameters, Dryad, Dataset, https://doi.org/10.5061/dryad.1g1jwstz2 [42].

Authors' contributions. X.Z.: conceptualization, data curation, formal analysis, investigation, methodology, validation, visualization and writing—original draft; Z.Z.: data curation, project administration, supervision and writing—

review and editing; C.Z.: project administration, resources, supervision and validation. All authors gave final approval for publication and agreed to be held accountable for the work performed therein.

Competing interests. The authors declare no conflict of interest.

Funding. The study was supported by the National Natural Science Foundation of China (nos. 41404084 and 41504094), the Natural Science Foundation of Hubei Province (no. 2013CFB396) and the National Science and Technology Major Project (no. 2017ZX05032003-005). We are also very grateful to Yangtze University for its support, including the Open Fund of Key Laboratory of Exploration Technologies for Oil and Gas Resources, Ministry of Education (K2021-08, K2021-03) and the Excellent Doctoral and Master's Degree Thesis Cultivation Program.

Acknowledgements. Many thanks to the editor of Royal Society Open Science and the anonymous reviewers. Your suggestions have improved the quality of the manuscript.

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
