## [Peer Review File · Royal Society Open Science]

Review History

RSOS-202122.R0 (Original submission)

Review form: Reviewer 1

Is the manuscript scientifically sound in its present form?

Yes

Are the interpretations and conclusions justified by the results?

Yes

Is the language acceptable?

Yes

Do you have any ethical concerns with this paper?

No

Have you any concerns about statistical analyses in this paper?

No

Recommendation?

Accept with minor revision (please list in comments)

Comments to the Author(s)

This paper discusses diagenetic facies classification into good and bad types, which is helpful for reservoir evaluation. Through comparing with logs can extrapolate to crosswell correlation.

This work is well written and can be recommended after some revisions.

1. There are too many figures in the manuscript, not sure if these are necessary, does these add to the paper? For example, figure 1 is cited from the other manuscript, it is not necessary to display again.
2. Introduction: More recent references should be added.
3. Figure 2 need scales.
4. core casting thin section-thin section, please clarify.
5. In part 4.1. Tight sandstones: texture, composition and diagenetic facies, use the word lithic fragments instead of cuttings.
6. Cut down some redundant parts in Conclusion.

Review form: Reviewer 2

Is the manuscript scientifically sound in its present form?

Yes

Are the interpretations and conclusions justified by the results?

Yes

Is the language acceptable?

Yes

Do you have any ethical concerns with this paper?

No

Have you any concerns about statistical analyses in this paper?

Yes

Recommendation?

Major revision is needed (please make suggestions in comments)

Comments to the Author(s)

In Figure 2 and text of manuscript, rock fragments should be used instead of cuttings.
For Figure 12, the amount of quartz, feldspar and rock fragments in each of the studied samples should be presented as a table.

In Figure 14, rock fragments should be used instead of cuttings and quartz secondary overgrowth is correct.

In classification of sandstones, refer to the Folk (1980) classification.

In Figure 4e, the opaque minerals are not in the form of cement but in the form of grains?
How is iron calcite diagnosed in Figure 4f? If the staining method is used, it should be mentioned in the study method.

The paragenetic sequence of these sandstones must be re-edited in the text with the figure. I'm a sedimentologist, that's better, a Petroleum Engineering Specialist should review this manuscript.

Review form: Reviewer 3

Is the manuscript scientifically sound in its present form?

No

Are the interpretations and conclusions justified by the results?

No

Is the language acceptable?

No

Do you have any ethical concerns with this paper?

No

Have you any concerns about statistical analyses in this paper?

No

Recommendation?

Major revision is needed (please make suggestions in comments)

Comments to the Author(s)

Thank you for the opportunity to review this paper, titled "Comparative study and discussion of diagenetic facies and conductivity characteristics based on experiments". Overall, this manuscript can contribute to our understanding of tight sandstone reservoirs and their petrophysical and electrical properties. Therefore, I recommend a major revision (or reject and resubmit). As it currently stands, however, the manuscript is far from ready for publication and requires major revision throughout the manuscript, particularly in their discussion. Although I found some of their findings are interesting, the authors need to provide a stronger justification of their interpretation and a much better justification of their diagenetic facies classification. Please find the annotated PDF file (Appendix A) for more detailed comments

Decision letter (RSOS-202122.R0)

Dear Dr Zhang

The Editors assigned to your paper RSOS-202122 "Comparative study and discussion of diagenetic facies and conductivity characteristics based on experiments" have now received comments from reviewers and would like you to revise the paper in accordance with the reviewer comments and any comments from the Editors. Please note this decision does not guarantee eventual acceptance.

Please submit your revised manuscript and required files (see below) no later than 21 days from today's (ie 14-Jun-2021) date. Note: the ScholarOne system will 'lock' if submission of the revision is attempted 21 or more days after the deadline. If you do not think you will be able to meet this deadline please contact the editorial office immediately.

on behalf of Professor Peter Haynes (Subject Editor)
openscience@royalsociety.org

Associate Editor Comments to Author:

We're sorry that it has taken an unusual length of time to complete the review of your work: a larger than average number of referees had to be approached to secure sufficient commentary for the editors to feel reasonably confident rendering a decision. Given the concerns of the referees, in particular reviewer #3, regarding over-interpretation of your data (including statements that do not appear to be supported by much evidence, from either the study itself or existing literature references) and also a range of linguistic concerns, we recommend revision. You may benefit from seeking language editing advice from a professional language editor (examples of which may be found at <https://royalsociety.org/journals/authors/benefits/language-editing/>, including a number the Royal Society has negotiated an author discount at) before resubmitting, too. The revision will be returned to the reviewers for further assessment, and the journal does not routinely permit multiple rounds of revision, so if you are not able to satisfy the reviewers and editors the work is ready for publication after revision, we may not be able to consider the paper further - we urge you to engage carefully with the reviewers' concerns in your tracked changes and point-by-point response document. Good luck!

Reviewer comments to Author:

Reviewer: 1

Comments to the Author(s)

This paper discusses diagenetic facies classification into good and bad types, which is helpful for reservoir evaluation. Through comparing with logs can extrapolate to crosswell correlation.

This work is well written and can be recommended after some revisions.

1. There are too many figures in the manuscript, not sure if these are necessary, does these add to the paper? For example, figure 1 is cited from the other manuscript, it is not necessary to display again.
2. Introduction: More recent references should be added.
3. Figure 2 need scales.
4. core casting thin section-thin section, please clarify.
5. In part 4.1. Tight sandstones: texture, composition and diagenetic facies, use the word lithic fragments instead of cuttings.
6. Cut down some redundant parts in Conclusion.

Reviewer: 2

Comments to the Author(s)

In Figure 2 and text of manuscript, rock fragments should be used instead of cuttings.

For Figure 12, the amount of quartz, feldspar and rock fragments in each of the studied samples should be presented as a table.

In Figure 14, rock fragments should be used instead of cuttings and quartz secondary overgrowth is correct.

In classification of sandstones, refer to the Folk (1980) classification.

In Figure 4e, the opaque minerals are not in the form of cement but in the form of grains?

How is iron calcite diagnosed in Figure 4f? If the staining method is used, it should be mentioned in the study method.

The paragenetic sequence of these sandstones must be re-edited in the text with the figure.

I'm a sedimentologist, that's better, a Petroleum Engineering Specialist should review this manuscript.

Reviewer: 3

Comments to the Author(s)

Thank you for the opportunity to review this paper, titled "Comparative study and discussion of diagenetic facies and conductivity characteristics based on experiments". Overall, this manuscript can contribute to our understanding of tight sandstone reservoirs and their petrophysical and electrical properties. Therefore, I recommend a major revision (or reject and resubmit). As it currently stands, however, the manuscript is far from ready for publication and requires major revision throughout the manuscript, particularly in their discussion. Although I found some of their findings are interesting, the authors need to provide a stronger justification of their interpretation and a much better justification of their diagenetic facies classification. Please find the annotated PDF file for more detailed comments (attached).

===PREPARING YOUR MANUSCRIPT===

one version identifying all the changes that have been made (for instance, in coloured highlight, in bold text, or tracked changes);
 a 'clean' version of the new manuscript that incorporates the changes made, but does not highlight them. This version will be used for typesetting if your manuscript is accepted.

===PREPARING YOUR REVISION IN SCHOLARONE===

- Any electronic supplementary material (ESM).
- If you are requesting a discretionary waiver for the article processing charge, the waiver form must be included at this step.
- If you are providing image files for potential cover images, please upload these at this step, and inform the editorial office you have done so. You must hold the copyright to any image provided.
- A copy of your point-by-point response to referees and Editors. This will expedite the preparation of your proof.

- Ensure that your data access statement meets the requirements at <https://royalsociety.org/journals/authors/author-guidelines/#data>. You should ensure that you cite the dataset in your reference list. If you have deposited data etc in the Dryad repository, please include both the 'For publication' link and 'For review' link at this stage.
- If you are requesting an article processing charge waiver, you must select the relevant waiver option (if requesting a discretionary waiver, the form should have been uploaded at Step 3 'File upload' above).
- If you have uploaded ESM files, please ensure you follow the guidance at <https://royalsociety.org/journals/authors/author-guidelines/#supplementary-material> to include a suitable title and informative caption. An example of appropriate titling and captioning may be found at https://figshare.com/articles/Table_S2_from_Is_there_a_trade-off_between_peak_performance_and_performance_breadth_across_temperatures_for_aerobic_scope_in_teleost_fishes_/3843624.

Author's Response to Decision Letter for (RSOS-202122.R0)

See Appendix B.

RSOS-202122.R1 (Revision)

Review form: Reviewer 1

Is the manuscript scientifically sound in its present form?

Yes

Are the interpretations and conclusions justified by the results?

Yes

Is the language acceptable?

Yes

Do you have any ethical concerns with this paper?

No

Have you any concerns about statistical analyses in this paper?

No

Recommendation?

Accept as is

Comments to the Author(s)

fine.

Review form: Reviewer 4

Is the manuscript scientifically sound in its present form?

Yes

Are the interpretations and conclusions justified by the results?

Yes

Is the language acceptable?

Yes

Do you have any ethical concerns with this paper?

No

Have you any concerns about statistical analyses in this paper?

No

Recommendation?

Accept with minor revision (please list in comments)

Comments to the Author(s)

Minor revision is needed

1. The ms have been revised and i AGREE WITH THE replies The authors provided
2. Improve the English grammar
3. remove some redundant parts in the revised manuscript
4. Improve the quality of some figures
, for instance ,add the scale in the thin sections and sem images

Decision letter (RSOS-202122.R1)

Dear Dr Zhang

On behalf of the Editors, we are pleased to inform you that your Manuscript RSOS-202122.R1 "Comparative study and discussion of diagenetic facies and conductivity characteristics based on

experiments" has been accepted for publication in Royal Society Open Science subject to minor revision in accordance with the referees' reports. Please find the referees' comments along with any feedback from the Editors below my signature.

You will see that Reviewer 4 is recommending further improvement to the paper and, as noted by the Associate Editor, the paper will not be accepted unless these comments are addressed (or satisfactory responses are provided).

The Reviewer's recommendations are:

(i) Improve the English grammar

(Please note the instructions below in the paragraph beginning 'If you have been asked to revise the written English ...'.)

(ii) remove some redundant parts in the revised manuscript

(This refers to the instructions to authors, e.g. at the top of p4 'The main text of the article should be ...' -- please check the whole paper carefully for this.)

(iii) Improve the quality of some figures, for instance ,add the scale in the thin sections and SEM images

(Figure 1 is very poor quality -- what is being shown is very unclear and the text is at very low resolution, Figure 3 lacks axes/colour scales etc -- please consider all Figures for intelligibility and legibility.)

Please submit your revised manuscript and required files (see below) no later than 7 days from today's (ie 05-Jan-2022) date. Note: the ScholarOne system will 'lock' if submission of the revision is attempted 7 or more days after the deadline. If you do not think you will be able to meet this deadline please contact the editorial office immediately.

on behalf of Peter Haynes (Subject Editor)
openscience@royalsociety.org

Associate Editor Comments to Author:

Comments to the Author:

The authors have made good efforts to respond to the reviewers' concerns, though a number of minor matters remain - that said, they do need to be addressed and we will not be able to accept the paper without them being tackled. Please ensure you fully respond to the reviewers' comments in your revised paper and make clear the changes made, and in a point-by-point response, too.

Reviewer comments to Author:

Reviewer: 1

Comments to the Author(s)

fine.

Reviewer: 4

Comments to the Author(s)

Minor revision is needed

1. The ms have been revised and i AGREE WITH THE replies The authors provided
2. Improve the English grammar
3. remove some redundant parts in the revised manuscript
4. Improve the quality of some figures
, for instance ,add the scale in the thin sections and sem images

===PREPARING YOUR MANUSCRIPT===

one version should clearly identify all the changes that have been made (for instance, in coloured highlight, in bold text, or tracked changes);

If you have been asked to revise the written English in your submission as a condition of publication, you must do so, and you are expected to provide evidence that you have received language editing support. The journal would prefer that you use a professional language editing service and provide a certificate of editing, but a signed letter from a colleague who is a proficient

user of English is acceptable. Note the journal has arranged a number of discounts for authors using professional language editing services (<https://royalsociety.org/journals/authors/benefits/language-editing/>).

===PREPARING YOUR REVISION IN SCHOLARONE===

-- If you are requesting an article processing charge waiver, you must select the relevant waiver option (if requesting a discretionary waiver, the form should have been uploaded, see 'File upload' above).

-- If you have uploaded any electronic supplementary (ESM) files, please ensure you follow the guidance at <https://royalsociety.org/journals/authors/author-guidelines/#supplementary->

material to include a suitable title and informative caption. An example of appropriate titling and captioning may be found at https://figshare.com/articles/Table_S2_from_Is_there_a_trade-off_between_peak_performance_and_performance_breadth_across_temperatures_for_aerobic_scope_in_teleost_fishes_/3843624.

Author's Response to Decision Letter for (RSOS-202122.R1)

See Appendix C.

Decision letter (RSOS-202122.R2)

Dear Dr Zhang,

I am pleased to inform you that your manuscript entitled "Comparative study and discussion of diagenetic facies and conductivity characteristics based on experiments" is now accepted for publication in Royal Society Open Science.

on behalf of Prof Peter Haynes (Subject Editor)
openscience@royalsociety.org

Appendix A**ROYAL SOCIETY
OPEN SCIENCE****Comparative study and discussion of diagenetic facies and
conductivity characteristics based on experiments**

Journal:	Royal Society Open Science
Manuscript ID	RSOS-202122
Article Type:	Research
Date Submitted by the Author:	10-Dec-2020
Complete List of Authors:	Zhou, Xueqing; Ministry of Education, Key Laboratory of Exploration Technologies for Oil and Gas Resources, Ministry of Education; Yangtze University Zhang, Zhansong; Ministry of Education, Key Laboratory of Exploration Technologies for Oil and Gas Resources, Ministry of Education; Yangtze University Zhang, Chaomo; Ministry of Education, Key Laboratory of Exploration Technologies for Oil and Gas Resources, Ministry of Education,; Yangtze University
Subject:	Geology < EARTH SCIENCES, Geophysics < EARTH SCIENCES, Petrology < EARTH SCIENCES
Keywords:	Diagenesis, water saturation, conductivity characteristics, rock cores experiments, electrical property
Subject Category:	Earth and Environmental Science

Author-supplied statements

Relevant information will appear here if provided.

Ethics

Does your article include research that required ethical approval or permits?:

This article does not present research with ethical considerations

Statement (if applicable):

CUST_IF_YES_ETHICS :No data available.

Data

It is a condition of publication that data, code and materials supporting your paper are made publicly available. Does your paper present new data?:

Yes

Statement (if applicable):

<https://doi.org/10.5061/dryad.vmcvdncrp>

<https://datadryad.org/stash/share/yHEPebLk8gy-vaFYn1KzEmLmlipxLmx1zIOLdsqqRtl>.

Conflict of interest

I/We declare we have no competing interests

Statement (if applicable):

CUST_STATE_CONFLICT :No data available.

Authors' contributions

This paper has multiple authors and our individual contributions were as below

Statement (if applicable):

Conceptualization, Xueqing Zhou; methodology, Xueqing Zhou; validation, Xueqing Zhou and Chaomo Zhang; formal analysis, Xueqing Zhou; investigation, Xueqing Zhou; resources, Chaomo Zhang; data curation, Xueqing Zhou and Zhansong Zhang; writingâ€”original draft preparation, Xueqing Zhou ; writingâ€”review and editing, Zhansong Zhang; visualization, Xueqing Zhou ; supervision, Zhansong Zhang and Chaomo Zhang; project administration, Zhansong Zhang and Chaomo Zhang. All authors have read and agreed to the published version of the manuscript.

Comparative study and discussion of diagenetic facies and conductivity characteristics based on experiments

Xueqing Zhou^{1,2}, Zhansong Zhang^{1,2,*} and Chaomo Zhang^{1,2}

1 Key Laboratory of Exploration Technologies for Oil and Gas Resources, Ministry of Education, Wuhan 430100, China

2 Hubei Cooperative Innovation Center of Unconventional Oil and Gas, Yangtze University, Wuhan 430100, China

Keywords: Diagenesis; water saturation; conductivity characteristics; rock cores experiments; electrical property

Summary

Tight gas reservoir in LX block has been considered as a critical field for the finding of oil and gas resources. The reservoir generally has low porosity and permeability, complex pore system, strong microscopic heterogeneity. Core samples of tight gas sandstone from the Ordos Basin were tested by using scanning electron microscopy, conventional core physical property analysis, core casting thin sections, core mercury intrusion experiments, and rock and electrical experiments; accordingly, diagenesis, diagenetic environment and diagenetic stage of the study block were examined. On this basis, the conductivity characteristics and saturation models of different diagenetic facies within the study area were systematically discussed for the first time. A number of experiments have shown that according to type of diagenesis, the pore structure, and the influence of the reservoir, a classification scheme for diagenetic facies (consisting of construction, cementation, destruction facies) can be set up. According to the diagenetic origin, the pore structure, pore type characteristics and other properties of various diagenetic facies were determined, and theoretical pore structure models of the three diagenetic facies were established. Based on these theoretical pore structure models, the construction facies was evaluated with a pore-connected vug-type conductivity model; the destruction facies was evaluated with a non-connected matrix pore conductivity model; and the cementation facies was evaluated with a residual intergranular pore conductivity model. Then, the rationality of each model and the effects of the parameters in each model on the final cementation exponent were analysed by simulation. These theoretical conductivity models further reveal the conductivity characteristics of reservoirs with complex pore structures and can guide the qualitative understanding of these characteristics in such reservoirs.

*Author for correspondence (zhangzhs@yangtzeu.edu.cn).

†Present address: Key Laboratory of Exploration Technologies for Oil and Gas Resources, Ministry of Education, Wuhan 430100, China

1. Introduction

Tight gas reservoirs are one of the unconventional resources that have been successfully developed around the world. Because tight sandstone reservoirs have experienced relatively strong diagenesis, they are generally characterized by variable lithologies, complex pore structures, diverse pore types, highly developed secondary pores, large pore size differences and strong heterogeneity [1,2]. The conductivity law of a logging curve is complex [3], resulting in tight sandstones that exhibit "non-Archie" behaviour [4]. In order to study the complex conductivity properties of tight sandstone reservoirs, scholars have evaluated the saturation of low-permeability sandstone reservoirs [5]. Two main ideas have emerged from this research. First, saturation models are based on a single controlling factor [6-10]. This principle originates mostly from analysing the influencing factors of rock electrical parameters, determining the dominant factors governing the conductive characteristics of rock, and establishing the corresponding conductive model, including physical experiments and numerical simulations revealed the influence of single factors on the electrical conductivity of rocks in the reservoir [11,12]. Many scholars studied the influence of shale content, pore structure and type and other factors on rock electrical parameters according to the existing research, the effect of individual factors (such as the shale content and type, pore structure, and pore type) on rock current conduction are deeply discussed. Second, saturation models are based on classifications [13,14].

However, the rock electrical parameters of tight sandstone reservoirs are usually controlled by multiple factors, and therefore, establishing a reasonable interpretation model by analysing only individual factors is difficult. For complex reservoirs with variable lithologies, complex pore structures, diverse pore types, the development of secondary porosity, large pore size differences, and considerable heterogeneity, the key is to investigate fundamental cause of the complexity of the reservoirs [15]. Accordingly, it is essential to identify classification units that can comprehensively reflect the reservoir complexities, determine their subdivisions, and approximate their properties. Several methods have been developed to subdivide the classification units of sandstone reservoirs; these methods are summarized into 2 types. The first type of classification method is an empirical classification technique based on three categories of petrophysical differences: (1) pore structure parameters [16-20], (2) Hydraulic flow units [21-24], and (3) a functional classification [25,26]. The other type of classification method is based on differences in geological origin [27], including lithofacies, sedimentary microfacies and diagenetic facies. Among them, diagenetic facies describe the comprehensive characteristics of present-day reservoirs by reflecting the properties, types of the reservoirs. The research on diagenetic facies is of broad significance for predicting and evaluating high quality reservoirs [28,29], the controls of physical parameters [30] and the effects of certain processes on microscopic pore throat structures and pore types. Certain achievements have been reported from other perspectives on this subject. Nevertheless, the research on the diagenetic facies told us that it may be able to comprehensively reflect the complex characteristics of the reservoir and can enable the calculation of saturation at a high accuracy.

Therefore, to represent reservoirs with complicated pore structures or rock properties strongly influenced by diagenesis, which leads to complex conductivity characteristics, take a tight gas sandstone reservoir in LX block in the northwestern Ordos Basin as an example. First, on the basis of comprehensive core casting thin sections, physical property results, rock and electrical experiments, and logging data, the diagenetic facies in LX block are classified. Second, the mechanisms responsible for controlling the formation and conductivity of diagenetic facies is studied. Then, the saturations of different diagenetic facies are established by theoretical derivation. The resulting model is based on a thorough understanding of the formation mechanisms and conductivity characteristics of reservoirs and can assist in evaluating the saturation of a complex reservoir that has experienced strong diagenesis.

The main text of your article should be split into informative sections. Usually these will be introduction, methods, results, discussion and conclusions; however, please feel free to use whatever headings and subheadings best suit your article. Abbreviations should be written out in full on first use.

2. Geological Background

2.1. Basin evolution

The Ordos Basin, which is a relatively simple yet large cratonic sedimentary basin, is located on the western edge of the North China Plate. The overall outline of the basin is rectangular. According to its structure and evolution, the Ordos Basin is divided into six first-order tectonic units (Figure 1). The study area is found within a structural unit of the generally monoclinic Jinxi flexure, and the central eastern region is an uplifted zone affected by the tectonic activity of the Zijinshan mineral field. The regional topography of LX block, which influenced by tectonic evolution in Ordos Basin, is high in the east and low in the west. The two main factors that led to the formation of the tight sandstones in the study area with unique geological features are sedimentation and diagenesis [31,32]. The resulting reservoir has the characteristics of low porosity and permeability (<0.1 mD).

2.2. Stratigraphy and depositional facies

In the upper Permian layers of the Ordos Basin, sedimentary microfacies, such as water distributary channel microfacies, underwater distributary channel microfacies and estuary bar microfacies, are mainly developed, and most oil and gas productivity occurs within these three main sedimentary microfacies; in the lower Permian layers, tidal flat and sand flat sedimentary microfacies are developed. Distributary channel deposits represent the underwater extensions of distributary channels in a delta plain. Sand and silt are the main types of sediment, and minimal shale is present. The underwater distributary channel deposits are generally dominated by sandstone and conglomerate-bearing sandstone; the conglomerate is relatively thin and fine-grained, and its sequence is characterized by a scouring surface at the bottom. An estuary dam is located in the estuary of the underwater distributary channels. Horizontal bedding and wavy bedding are present in the lower part of the channel, and parallel bedding appears at the top. In addition, slump deformation bedding and drainage structures are well developed. Tidal flat and sand flat sedimentary microfacies are mostly developed near the coastline and are divided into sand flat, mud flat and mixed flat microfacies. The main gas reservoirs in LX block are concentrated in the upper Palaeozoic, namely, the Permian, and are divided into 6 groups, which constitute the main research objects of this paper [34].

3. Materials and Experiments

3.1. Materials

In this study, 18 tight sandstone cores were collected from 5 wells in the Permian section of the upper Palaeozoic in LX block (Figure 1). The data were obtained from scanning electron microscopy (SEM), core casting thin sections, conventional core physical property experiments, core mercury intrusion experiments, organic geochemical analyses, rock and electrical experiments. For rock and electrical experiments, a PLS-200 rock electrical conductivity meter was adopted to measure growth; a small plunger with a radius of 3.5 cm was used in order to make sure the accuracy of the experiment. The experiment was carried out with a NaCl solution with a salinity of 20 g/L under a confining pressure of 5 MPa at room temperature. The core casting thin sections were obtained in accordance with the SY/T 5368-2000 Chinese national standard [35] under polarized light at room temperature. A JSM-5500LV scanning electron microscope was used for the microscopic observations and acquisition of representative photographs, and experimental measurements were carried out in an environment with 40% humidity that was maintained at 25°C.

3.2. Experiments

To determine the mineral and rock compositions of the 18 rock samples, representative and fresh sections of the samples were selected to prepare core casting thin sections and SEM specimens. Polarization microscopy and SEM were used for microscopic observations. The core casting thin sections were examined in accordance with the SY/T 5368-2000 Chinese national standard [35], and the SEM experiment was conducted in accordance with the SY/T 5162-1997 Chinese national standard. Panoramic and local images were obtained at different magnification ratios in conjunction with sample analyses and thin section identification. The rock electrical parameters of the 18 rock samples were obtained in accordance with the SY/T 5385-2007 Chinese national standard; the environmental temperature and humidity during the experiment were 25 °C and 35-50% RH, respectively. Eighteen plunger samples were drilled perpendicular to the drilling core column, and the length was approximately 35 mm and diameter was approximately 25 mm. The rock samples were saturated with a NaCl solution, and the resistivities corresponding to different saturation amounts were measured to obtain the rock electrical parameters. Finally, the capillary pressure curves of the rock samples were obtained by mercury injection. Referring to the SY/T 5346-2005 Chinese national standard ("Determination of rock capillary pressure curves of rock"), the rock samples were cleaned and dried to constant weight, and their geometric dimensions, porosity and air permeability were measured. Then, the core chamber valve was opened, the mercury supplement valve was opened, and the height of the mercury cup was adjusted. The distance between the liquid level of the mercury cup and the suction valve corresponded to the height of the mercury column (approximately 760 mm) under the current atmospheric pressure; then, the isolation valve was opened, and the height of the mercury cup was adjusted again. At this time, the output value of the differential pressure sensor was between 28.00 and 35.00 cm; the suction valve was closed, the vacuum pump was closed, the vacuum pump was opened, and the mercury supplement valve was closed. The inlet valve of the high-pressure metering pump was turned off, and the metering pump was adjusted so that the minimum gauge pressure was zero. According to the set pressure, the pressure and height of the mercury column in the mercury volume measuring tube after stabilization were recorded step by step until the highest set pressure was reached; the same procedure was conducted in reverse until the lowest set pressure was reached. Finally, according to the experimental results, the experimental data were sorted.

4. Results

4.1. Tight sandstones: texture, composition and diagenetic facies

The overall analysis based on the core casting thin section and SEM observations and statistics for the specimens from the study area, it is found that the main rock minerals is quartz, followed by cuttings (Figure 2). The cuttings are abundant with a complex composition; they are composed mainly of volcanic rocks and volcanoclastic rocks but include some metamorphic rock debris and sedimentary rock debris. The main lithologies in LX block are lithic sandstone and lithic quartz sandstone with minor quartz sandstone. According to the interstitial and rock structures, the study area has experienced strong diagenesis; the compositions of the interstitial materials are diverse and complex, the reservoir is dense, and the particle size distribution is wide and uneven, reflecting a complicated geological history. The main pore types are secondary pores, including dissolution intergranular pores, intragranular dissolution pores, and residual intergranular pores. Within the primary pores, intergranular pores, intragranular dissolved pores, intercrystalline pores, and fractured pores are all developed.

In addition to identifying the mineral components of the rock samples, the diagenetic facies of 18 rock samples were determined by observing the main diagenesis and pore characteristics of each rock sample under a microscope. To clarify the process employed to determine the diagenetic facies in more detail, the definition of diagenetic facies and their existing classification schemes are briefly introduced, as is the method used to classify the diagenetic facies in LX block. Then, the physical properties, characteristics of pore, texture and composition of the various diagenetic facies within the reservoir are analysed and summarized.

This section may be divided by subheadings. It should provide a concise and precise description of the experimental results, their interpretation as well as the experimental conclusions that can be drawn.

4.1.1. Definition of diagenetic facies

Diagenetic facies, which include rock particles, cement, structural features and pores and contain information regarding its comprehensive appearance and pore evolution, is the product of certain evolutionary stages under specific sedimentary environments and tectonic settings [36]. The concept of diagenetic facies was proposed as early as 1968; the study of diagenesis has since experienced breakthroughs up to 1990 and remains the focus of sedimentologists and research on oil and gas reservoirs. Following decades of exploration and development, diagenetic facies can be summarized as representations of the comprehensive characteristics of present-day reservoirs and can provide genetic indicators of reservoir properties and reservoir types and their advantages and disadvantages. According to previous studies and the definition of diagenetic facies, diagenetic facies can be classified according to the following three features: diagenesis, diagenetic environment and diagenetic stage [37]. The diagenetic environment refers to the temperature, pressure and redox conditions of the sediments during diagenesis. Diagenetic products contain two types of materials: minerals that pre-date diagenesis and minerals that were newly produced during diagenesis. These materials, such as secondary growth rims in quartz, authigenic chlorite, carbonate minerals, and kaolinite, can play indicative roles in diagenetic facies. Diagenetic facies classification schemes refer to other features, including the diagenetic stage and evolutionary sequence of events, the diagenetic facies, and the diagenetic system. Thus, the main factor controlling the diagenetic facies of clastic rock consists of diagenesis (that is, the diagenetic environment and diagenetic products), and the diagenetic properties influence the corresponding classification of diagenetic facies. As a result of the complexity involved in this process, to date, no unified and clear standard for the classification of diagenetic facies has been proposed. In other words, the classification results for this study must satisfy and provide a useful research foundation for subsequent studies.

4.1.2. Classification of diagenetic facies

The diagenesis and diagenetic environment, stage and products in the study area are analysed in this section according to the definition and classification of diagenetic facies. According to the previous studies [38], combined with the research work, diagenetic sequence and stage of the reservoirs in LX block are shown in Figure 3, the reservoir is typified by the middle of diagenetic stage B, which is typically characterized by early chlorite rims and late ferric calcite cementation, indicating that diagenesis environment of the study area was acidic. Thus, due to the consistency of the diagenetic environment and all in stage B, the diagenetic facies can be classified by differences in diagenetic properties across the study area, thereby providing a reasonable classification scheme for subsequent research on the lithoelectrical relationship and electrical conductivity characteristics. Accordingly, diagenetic facies are named according to the effects of increasing or decreasing reservoir capacity from a practical exploration perspective [37]. Diagenetic facies are used mainly to represent the diagenetic processes that control the physical properties of different facies. The diagenetic facies of the study area were identified mainly through multiscale analytical and experimental data and the experimental results, particularly through thin section observations, SEM, particle size distributions, X-ray diffraction clay content analysis, mercury injection, nuclear magnetic resonance and calculations of the over pressured pore permeability [39]. There are three types of diagenetic facies: construction facies, cementation facies, destruction facies. As follows, we introduce specific characteristics for each diagenetic facies.

The construction facies is the best reservoir type for hydrocarbon accumulation in LX block. The main diagenetic processes responsible for this facies are dissolution and the precipitation of authigenic chlorite. On the one hand, in an acidic diagenetic environment, potassium feldspar and albite react with H₂O and carbon dioxide or react with organic acids (see Eq. 1 to Eq. 5). The resulting dissolution is strong, leading to the formation of dissolution pores, which become the most important type of storage space in the construction facies (Figure 4(a) ~ Figure 4(b)). On the other hand, in the early stage of diagenesis, montmorillonite is converted into chlorite under iron-rich and manganese-rich conditions; the content of

montmorillonite, which is mainly due to the precipitation of authigenic chlorite, is positively related to the physical properties of the reservoir. Chlorite exists mainly in the form of granular films (pore linings). Many studies have shown that chlorite films (Figure 4(c)) that form around minerals such as quartz can effectively inhibit the growth of the mineral; the growth of the film around the edges is increased, and its presence enhances the rock's resistance to compaction, which is greatly important for protecting the pore space.

The cementation facies includes both siliceous and carbonate cements; between them, however, because siliceous cementation occurs after the formation of the chlorite film and is inhibited to some extent, the cementation facies consists mainly of carbonate cement. According to the main diagenetic products in the cementation facies, siliceous and carbonate cementitious materials can effectively enhance the compaction resistance of the reservoir to some extent. However, these two diagenetic products mainly occupy pores. In terms of the siliceous cement, the secondary growth of quartz is dominant, and quartz grows outwardly from the grain surfaces (Figure 4(d)), which reduces the pore space of the reservoir to some extent and has a large impact on the physical properties. In contrast, the carbonate cement mainly fills pores. In several carbonate cementation zones, the pores are filled, and channels enabling fluid flow are blocked, greatly reducing the overall porosity and permeability. The degree to which carbonate cement is developed has important influences on the physical properties and heterogeneity of the reservoir. The main carbonate cement in LX block is composed of calcite, ferric calcite and some ferric dolomite; according to the staining characteristics observed from core casting thin sections, the carbonate cement in LX block comprises mainly early siderite, late calcite and minor iron-bearing dolomite (Figure 4(e) ~ Figure 4(f)). Early-stage cements consist of micritic calcite, usually without the metasomatism of clastic particles, and generally present a base-type or intergranular distribution. Such cementation plays an important role in enhancing the compact ability of sandy sediments, thereby preserving the intergranular volume and generating secondary pores for dissolution by acidic fluids in the later stage. Late-stage cements appear as fillers padding intergranular or dissolution pores, which is mainly present in mineral components like quartz, feldspathic, debris; the composition of these late cements mainly include ferrocalcite and minor ankerite, resulting in a relatively close line of contact between the particles and cement, which is usually composed of late-stage carbonate. This contact most commonly has a detrimental effect on the reservoir properties.

The main components of the destruction facies are precipitated kaolinite and sericite and partially precipitated authigenic illite. Kaolinite is widely distributed throughout LX block (Figure 4(g)) and exists mainly in intergranular or heterogeneous groups, residual intergranular pores or feldspathic debris dissolution holes; kaolinite mostly fills in granular dissolution holes, which results in a poor reservoir quality. The formation of kaolinite is caused mainly by the dissolution of feldspar (Eqs. 1-5) and dark minerals, which have the basic properties of volcanic rocks, during early diagenesis (Eqs. 6-10). Plagioclase provides the necessary aluminium ions for the development of kaolinite. According to SEM, the kaolinite

within the intergranular pores is scaly (Figure 4(h)). Kaolinite can be used as an important supply for illite (by being converted into illite) in a high-temperature environment. Illite mainly fills in pores exhibiting a fibrous appearance, and it has a relatively low abundance. Authigenic illite mainly refers to the illite formed during the diagenesis process after the sedimentation stage. According to several studies, the formation of illite according to the different sources can be summarized into 3 types: the conversion of montmorillonite, kaolinite or potassium feldspar into illite, which directly crystallizes. In the process of diagenesis, the conversion of montmorillonite into illite and chlorite often goes through a mixing stage, namely, the mixture of illite and montmorillonite and the mixture of illite and chlorite. Overall, the Taiyuan Formation and the Benxi Formation were deposited in a marine environment, in which feldspar was strongly corroded, resulting in the precipitation of illite. Self-generated illite is relatively well developed compared with that in the Stone Box Group and the Shiqianfeng Group; the burial depth is deep, and the formation temperature is high. This environment is also suitable for the conversion of kaolinite into illite. Due to the diagenetic environment of the destruction facies, the pores are mainly filled with pore fluids, which make a detrimental impact on the porosity and pore structure.

4.2. Tight sandstones: electrical characteristics

Through the rock and electrical experiments, we obtained rock electrical parameters for the 18 rock samples; the resulting relationship between formation factor (F) and porosity (Por) [40] is shown in Figure 5. The 18 rock samples have undergone intense diagenesis and are characterized by complex mineral compositions, complicated pore structures and diverse pore types with a wide porosity distribution. The straight lines in the Figure indicate the relationship between the formation factor and porosity ranging from the ratio factors $a=1$ and $m=1$ to the ratio factors $a=1$ and $m=2.6$. From the relationship between the formation factor and porosity, in the case of $a=1$, m varies greatly from 1.3 to 2.6. As shown in Figure 5, with decreasing porosity, the formation factors corresponding to low porosity deviate from the straight line of high porosity. The main reasons for this phenomenon are the increase in the shale content in small pores, the complexity of the structures and the increase in the bound water content. With 10% porosity as the boundary, different cementation exponent can be obtained for the high- and low-porosity parts. Within the two sections with high and low porosities, the relationship between F and Por exhibits somewhat evident non-Archie characteristics. According to the cross plot (Figure 6) between the resistivity (Rt) and the water saturation (Sw) of the 18 rock samples, different rock samples have different relationship between Rt and Sw, non-Archie characteristics are also observed in the relationship between the Rt and Sw.

4.3. Analysis of factors influencing the electrical conductivity and pore characteristics of the diagenetic facies

The factors influencing the electrical conductivity of the diagenetic facies were comprehensively analysed based on core casting thin sections and SEM, mercury injection and rock electrical experiments performed on the 18 rock samples. According to the rock electrical experiments, as mentioned above, the relationship between F and por in the study area does not conform to Archie's formula. Consequently, the individual

factors affecting the conductivity were analysed from the perspectives of the lithological composition, pore
structure and pore type. Among them, the component maturity and clay content are the two parameters
representing the lithological composition, and the average pore throat radius obtained by mercury
injection is chosen to characterize the pore structure (Figure 7(a) ~ (c)). On the whole, the cementation
exponent (m) is affected by the component maturity, clay content, and average pore throat radius, but no
good correlation is observed. To study the influence of the pore type on the rock electrical parameters, the
dual-pore medium model proposed by Rasmus [41] was adopted to understand the controlling effect of
the pore type on the electrical properties of rocks in tight gas reservoirs. According to thin section data,
the pore structure of a tight gas reservoir can be generalized into intergranular pores, intergranular
dissolved pores, residual intergranular pores, and clay residual intergranular-intergranular pores. The
relationships between the formation factors of different pore types and porosity (Figure 5) can be simulated
by the Rasmus model. Several pore types in Figure 5 correspond to regions 0-3. Figure 5 further shows that
the existence of dissolved pores increases the rock electrical parameter m ; the smaller the matrix porosity
is and the larger the dissolved pores are, the greater the increment in the rock electrical parameter m .
Compared with rocks that have intergranular pores with the same porosity, rocks with clay residual
intergranular-intergranular pores have a lower resistivity; that is, the rock electrical parameter m
decreases.

Because of the complexity of conductivity characteristics, rock electrical parameters are affected by
multiple factors. Therefore, it is necessary to find a suitable classification unit to simplify the conductivity.
According to previous studies in the Ordos Basin [33], strong diagenesis is responsible for the complex
structures, variety of pore types and complex lithological compositions of the units within the basin.
Previous studies have shown that diagenetic facies was a reasonable classification unit, the difference of
the pore system is one of the factors causing the conductivity characteristics. In addition, clay is also an
influencing factor. Therefore, lithoelectrical parameters of the 18 rock samples from different diagenetic
facies were analysed; the relationship between m and n for different diagenetic facies is shown in Figure
8. Overall, the m value of the construction facies shown in Figure 9 is higher than others. On the one hand,
the average m value of the construction facies conforms to the relationship between m and the average
pore radius r_{mean} illustrated in Figure 8. Because the average pore radius in the construction facies is
higher than others, and value of parameter m increases. On the other hand, the development of dissolution
pores in the construction facies and the increase in dissolution pores both increase the m value.

It is necessary to clarify the differences of electrical conductivity in lithoelectrical parameters of
diagenetic facies; that is, conductivity characteristics of the different diagenetic facies vary. To clarify the
electrical conductivity characteristics of the 3 diagenetic facies, first, the influencing factors of the electrical
conductivity of each diagenetic facies were analysed. According to a previous similar study in the Ordos
Basin [33], within each diagenetic facies, the lithoelectrical parameters are affected by a single main factor,
and m in LX block is controlled mainly by the pore characteristics. Differences in the pore structures among
diagenetic facies cause the parameter m of these diagenetic facies to vary. To better study the conductive
characteristics of diagenetic facies, we should first systematically do research on the pore characteristics of
those diagenetic facies and then conductivity models of the diagenetic facies can be established.

Porosity can reflect the reservoir capacity of a rock, while the permeability, pore throat size,
connectivity and particle size distribution determine the flow characteristics. Therefore, the pore size, pore
structure, pore type, interconnectivity and relationships between these factors of various diagenetic facies
must be studied to establish the pore structure models of those facies. First, according to physical property
data from the tight sandstone cores, the distributions of the porosity and permeability are shown in Figure
9. The porosity of the cementation facies varies between 2% and 10%, although it is generally concentrated
at 8%, and the permeability ranges between $0.2 \times 10^{-3} \mu\text{m}^2$ and $1 \times 10^{-3} \mu\text{m}^2$. The porosity of the destruction
facies varies between 2% and 9% with an average of 4%, and the permeability varies between $0.01 \times 10^{-3} \mu\text{m}^2$
and $0.2 \times 10^{-3} \mu\text{m}^2$ with an average of approximately $0.1 \times 10^{-3} \mu\text{m}^2$. The porosity of the construction facies

ranges between 5% and 16%, and most of the porosity is above 7%, and the permeability varies between
$0.1 \times 10^{-3} \mu\text{m}^2$ and $10 \times 10^{-3} \mu\text{m}^2$, with most permeability values exceeding $1 \times 10^{-3} \mu\text{m}^2$. The physical properties
of the construction facies are superior compared with others.

Next, the parameters that reflect the pore structures and pore types of various diagenetic facies were
calculated (Figure 10~13) according to the mercury injection and core casting thin section data. Figure 10
reveals that the degrees of mercury distortion of the three diagenetic facies are all distributed between -1
and 1. The values of r_{35} and r_{mean} for the cementation facies are smaller than those for the construction
facies, and the values of Pd and Sp are relatively low as well; most of the values are concentrated. These
mercury injection parameters indicate that the cementation facies physical properties are moderately good.
In contrast, among all diagenetic facies, the values of r_{35} and r_{mean} for the destruction facies are relatively
small, while the Pd value is relatively low, and the content of lithological debris among the rock
components is relatively high (Figure 12). For the construction facies, the degree of mercury distortion
(skp) of the construction facies is greater than 0, which is coarse; additionally, the values of r_{35} and r_{mean}
are relatively large, and that of Pd is relatively low (Figure 11 and 13), while the value of Sp is
comparatively high, and the quartz content is higher for the construction facies than for the other two
diagenetic facies (Figure 12).

Overall, the mercury injection parameters indicate that pore structures of the construction facies are
better than those of the other two diagenetic facies and that its pore distribution is relatively uneven.
Moreover, the pore space of the construction facies is composed of residual intergranular pores and
dissolution pores, and the overall pore structure is more complex. In contrast, for the cementation facies,
while the pore distribution is asymmetric, the pore space is dominated by residual intergranular pores,
and there is mostly a single pore type; in addition, the pore throat distribution is more concentrated, and
the overall pore structure is relatively simple. For the destruction facies, the pore distribution is relatively
asymmetrical, and its physical properties are poor; the pore space is composed mainly of residual
intergranular pores and small dissolved pores, and the pore types are relatively complicated.

According to pore characteristics of the diagenetic facies in LX block, schematic maps of the diagenetic
facies were drawn. The schematic map of the construction facies shown in Figure 14(a) was established
according to the core casting thin section analysis, mercury injection data and other experimental analysis
results. Dissolution is the main process in this facies, in addition, it also includes chlorite precipitation;
precipitated chlorite is the main cementing substance, and the clay cementing substances appear mainly
exist as linings, which greatly improve the anti-compactability of the reservoir and are beneficial to the
preservation of primary pores. Then, the intensity of dissolution is strong, leading to the formation of many
secondary pores and increasing both the reservoir space and the reservoir interpore connectivity. The
porosity is above 7%, most permeability values exceeding $1 \times 10^{-3} \mu\text{m}^2$ (Figure 10). Overall, construction
facies with relatively high porosity and permeability, also called sweet spots, are highly sought by
petroleum geologists. In contrast, the cementation facies (Figure 14(b)) is composed mainly of carbonate
cement and develops mostly in thick sandstone due to smooth fluid migration, and those in fine
sandstones experiencing early compaction, the physical properties and pore structures of the fine
sandstones are poor, as pore fluids cannot migrate effectively within the pores; therefore, the siliceous
cement and carbonate cement contents are lower in fine sandstones than in thick sandstones. Finally,
the schematic map of the destruction facies is shown in Figure 14(c). The main characteristics of the destruction
facies are sedimentation and filling with kaolinite and sericite. The development of a large amount of
kaolinite often signifies a reduction in primary porosity and the generation of a large number of secondary
dissolution pores. The reaction that precipitates sericite produces small-scale sediments that destroy the
porosity and pore throats. Hence, the main processes responsible for the destruction facies are the
autogenetic cementation of kaolinite and the sericitization of volcanic debris-filling pores.

5. Discussion and Deduction of the Conductivity Model

5.1. Conductivity characteristics

For reservoirs with the very complex pore system, so, the application of Archie's formula is limited; this restriction is reflected mainly in the significant differences among the pore index magnitudes of various types of reservoirs. In other words, the complexity of the pore structure intensifies the variations in the values of the cementation exponent m in the classic Archie formula, which affects the accuracy of the saturation calculation. According to the analysis and summary of physical properties and the pore structure characteristics of the various diagenetic facies within LX block, the three diagenetic facies clearly exhibit different pore structure characteristics [42]. Therefore, according to the characteristics of the pore structures of the three diagenetic facies, conductivity characteristics are discussed, and conductivity models suitable for these diagenetic facies are deduced.

5.1.1. Destruction facies

According to the thin section observations and mercury injection experiment, the destruction of clay residual intergranular pores and intergranular pores [43] is suitable for a conductivity model of this dual-pore system. This model, which can be abstracted for matrix pores, interconnecting system of porosity composed of intergranular pores (clay residual intergranular pores); further, this model can be used in abstract conductivity models [44]. These models assume that the destruction facies of subsurface rocks are completely saturated with water, and the rock resistivity R_{des} is calculated as follows:

$$R_{des} = v_{nc} \Phi R_w + (1 - v_{nc} \Phi) R_o. \quad (11)$$

When the pore structure type in the rock consists of only matrix pores,

$$F = \Phi_b^{-m_b}. \quad (12)$$

When the pore structure within the rock comprises composite pores,

$$R_{des} = F_t R_w \quad (13)$$

and

$$F_t = \Phi^{-m}. \quad (14)$$

Then, Eq. 13 can be substituted into Eq. 11:

$$F_t R_w = v_{nc} \Phi R_w + (1 - v_{nc} \Phi) R_o. \quad (15)$$

Dividing both sides of Eq. 15 by R_w :

$$\begin{aligned} F_t &= v_{nc} \Phi R_w + (1 - v_{nc} \Phi) R_o / R_w \\ &= v_{nc} \Phi + (1 - v_{nc} \Phi) F. \end{aligned} \quad (16)$$

Then, Eq. 12 and Eq. 14 can be substituted into Eq. 16:

$$\Phi^{-m} = v_{nc} \Phi + (1 - v_{nc} \Phi) \Phi_b^{-m_b}. \quad (17)$$

Next, the cementation exponent m of the destruction facies can be expressed as:

$$m = \frac{\log[v_{nc} \Phi + (1 - v_{nc} \Phi) \Phi_b^{-m_b}]}{-\log \Phi} \quad (18)$$

Lucia's vug porosity ratio can be expressed as:

$$v_{nc} = \frac{\Phi_{nc}}{\Phi} = \frac{\Phi - \Phi_m}{\Phi} = \frac{\Phi - \Phi_b}{\Phi(1 - \Phi_b)} \quad (19)$$

5.1.2. Construction facies

The overall reservoir conditions of the construction facies are superior to that of the destruction facies and the cementation facies. The pore throats are composed mainly of dissolved feldspar pores and intergranular pores, and the construction facies can be abstracted into a pore system composed of matrix pores and interconnected pores (dissolved pores). Abstracting the parallel conductivity model [41], the overall resistance r_{con} of the rock has the following relationship:

$$\frac{1}{r_{con}} = \frac{1}{r_b} + \frac{1}{r_d} \quad (20)$$

where

$$r = R \frac{L}{A} \quad (21)$$

Then, Eq. 20 can be expressed as:

$$\frac{A_{con}}{R_{con} L_{con}} = \frac{A_b}{R_b L_b} + \frac{A_d}{R_d L_d} \quad (22)$$

Multiplying both sides of Eq. 22 by L_{con}^2 yields:

$$\frac{A_{con} L_{con}}{R_{con}} = \frac{A_b L_{con}}{R_b} \frac{L_{con}}{L_b} + \frac{A_d L_{con}}{R_d} \frac{L_{con}}{L_d} \quad (23)$$

Here, we define the entire rock volume as 1 and $v_b = L_b / L_{con}$; then, Eq. 23 becomes:

$$\frac{1}{R_{con}} = \frac{A_b L_{con}}{R_b v_b} + \frac{A_d L_{con}}{R_d v_d} \quad (24)$$

where R_b can be expressed as $\frac{R_w}{\phi_b^M S_w^{Nb}}$. Eq. 24 can thus be expressed as:

$$\frac{1}{R_{con}} = \frac{V_b \phi_b^M S_w^{Nb}}{R_w} + \frac{V_d S_w^{Nd}}{R_w v_d^2} \quad (25)$$

V_d can be expressed as the difference between the total porosity and primary porosity:

$$V_d = \phi - \phi_b \quad (26)$$

$$V_b = 1 - V_d = 1 - (\phi - \phi_b) \quad (27)$$

Then, Eq. 26 and Eq. 27 can be substituted into Eq. 25:

$$\frac{[1 - (\varphi - \phi_b)] [\phi_b^{M_b} S_{wb}^{N_b}]}{R_w} + \frac{1}{R_{con}} = \frac{(\varphi - \phi_b) S_{wf}^{M_f} \left(\frac{1}{v_d^2}\right)}{R_w}. \quad (28)$$

When the rock is completely saturated with water, R_w can be expressed as R_o ; then,

$$\frac{R_w}{R_o} = [1 - (\varphi - \phi_b)] \phi_b^{M_b} + (\varphi - \phi_b) \frac{1}{v_d^2}, \quad (29)$$

where

$$\frac{R_w}{R_o} = \frac{1}{F'} = \varphi^{M'}. \quad (30)$$

Next, Eq. 30 can be substituted into Eq. 29:

$$M = \frac{\log \left\{ [1 - (\varphi - \phi_b)] \phi_b^{M_b} + \left[(\varphi - \phi_b) \frac{1}{v_d^2} \right] \right\}}{\log \varphi}. \quad (31)$$

Letting $\frac{1}{v_d^2} = 1$,

$$M = \frac{\log \left\{ [1 - (\varphi - \phi_b)] \phi_b^{M_b} + [(\varphi - \phi_b)] \right\}}{\log \varphi}. \quad (32)$$

5.1.3. Cementation facies

According to the previous analysis, the cementation facies are composed primarily of residual intergranular pores, and its pore types are relatively simple. From a pore structure perspective, the cementation facies pore throat distribution is relatively concentrated, and the overall pore structure is relatively simple. This analysis demonstrates that the cementation exponent of the cementation facies is controlled by (and has a close relationship with) the porosity (Figure 15).

$$m = 0.3959 * \ln(\varnothing) + 1.0586. \quad (33)$$

Where \varnothing is porosity.

5.2. Analysis of the rationality of the model and influencing factors

[revised manuscript text omitted]

18 core samples were determined based on microscopic observation of thin section observations, SEM, and pore structure experiments such as mercury injection, nuclear magnetic resonance and calculations of the over pressured pore permeability. Six diagenetic rock samples pertain to each of the construction facies, cementation facies and destruction facies. The parameters of other conductive models are also obtained by laboratory measurement. Among these samples, the porosity is obtained by physical property experiments; the matrix porosity, dissolution porosity and non-connected porosity are obtained by core casting thin sections; and the cementation exponent is obtained by rock and electrical experiments. The specific experimental data are shown in Table 1.

Using the corresponding conductivity models of the various diagenesis facies, the conductivity model of a dual-pore system is used for the destruction facies, and the construction facies can be abstracted into a pore system composed of matrix pores and interconnected pores (dissolved pores), the parallel conductivity model is used for the construction facies. The cementation facies with relatively simple pore type is evaluated by porosity- cementation exponent model. The cementation exponents are calculated by Eq. 18, Eq. 32, and Eq. 33 respectively. Then, compared with the cementation exponents measured in the laboratory, as shown in Figure 24. According to the evaluation results of the three diagenetic facies, the cementation exponent of each diagenetic facies are in good agreement with the experimental results.

6. Conclusions

For complex tight sandstone reservoirs that have been subjected to strong diagenesis and exhibit complex electrical conductivity characteristics, diagenetic facies is the suitable classification unit. The reservoir is typified by the middle of diagenetic stage B, diagenesis environment of the study area was acidic. Due to the consistency of the diagenetic environment and all in stage B, the diagenetic facies was classified by differences in diagenetic properties across the study area, and the reservoir is divided into 3 types of diagenetic facies: construction facies, cementation facies, destruction facies. Accordingly, the main diagenetic properties, pore sizes, pore structures, pore types, interconnectivities and relationships between these factors of various diagenetic facies are also studied. Then, using the conductivity model of a dual-pore system, the destruction facies is abstracted into a pore system consisting of matrix pores and non-connected pores (clay intercrystalline pores) that can be described using a series conductivity model. In contrast, the construction facies is abstracted into a pore system consisting of matrix pores and interconnected pores (dissolved pores), as determined by a parallel conductivity model. The pore structures and pore types of the cementation facies in the study area are simple, and the cementation exponent is controlled by the amount of porosity within the rock. A simulation method is adopted to examine the relationships between various diagenetic facies conductivity models and different types of porosity by means of intersection diagrams. For the destruction facies, when the number of non-connected pores in the system is large, the cementation exponent decreases as the total porosity increases. When the total porosity is small, namely, when non-connected pores occupy a relatively high proportion of the rock, the cementation exponent decreases rapidly. With an increase in the total porosity, the proportion of non-connected pores decreases, and the change in the m value becomes small. For the construction facies, when the dissolution porosity is small, the influence of the dissolution pores on the cementation exponent is very small. As the dissolution porosity increases, as the proportion of dissolution porosity increases, the cementation exponent decreases. The diagenetic facies conductivity models are reasonable and can characterize the conductivity characteristics of the various diagenetic facies.

7. Nomenclature

A_b = area of matrix

A_{con} = area of volume

A_d = area of interconnected pores

F = formation factor

F_t = formation factor of composite system

L_b = length of matrix

L_{con} = length of volume

L_d = length of interconnected pores

m_b = matrix cementation exponent

8 m = cementation exponent of destruction facies

M = cementation exponent of construction facies

R_{des} = resistivity of composite system (matrix pores and unconnected pores (clay intergranular pores)) at 100% saturated water

r_{con} = resistance of composite system (matrix pores and interconnected pores (dissolved pores))

r_b = resistance of matrix

r_d = resistance of interconnected pores

R_{con} = resistivity of composite system (matrix pores and interconnected pores (dissolved pores))

R_b = resistivity of matrix

R_d = resistivity of interconnected pores

R_w = formation water resistivity at formation temperature

R_o = resistivity at reservoir temperature when the 100% saturated resistivity is R_w , the resistivity of the matrix system

v_{nc} = ratio of non-connected pores

v_b = ratio of matrix pores

v_d = ratio of interconnected pores

φ = porosity

φ_b = matrix porosity related to the total volume of the matrix system

**Acknowledgments**

Many thanks to the editor of Royal Society Open Science and anonymous reviewers. Your suggestions have improved the quality of the manuscript.

**Funding Statement**

The study was supported by the National Natural Science Foundation of China (Nos. 41404084, 41504094), the Natural Science Foundation of Hubei Province (Nos. 2013CFB396), and the National Science and Technology Major Project (Nos. 2017ZX05032003-005). And also very grateful to Yangtze University for its support, including the Open Fund of Key

Laboratory of Exploration Technologies for Oil and Gas Resources, Ministry of Education (K2016-09, K2017-01, K2018-11), and the Excellent Doctoral and Master's Degree Thesis Cultivation Program.

Competing Interests

The authors declare no conflict of interest.

Authors' Contributions

Conceptualization, Xueqing Zhou; methodology, Xueqing Zhou; validation, Xueqing Zhou and Chaomo Zhang; formal analysis, Xueqing Zhou; investigation, Xueqing Zhou; resources, Chaomo Zhang; data curation, Xueqing Zhou and Zhansong Zhang; writing—original draft preparation, Xueqing Zhou ; writing—review and editing, Zhansong Zhang; visualization, Xueqing Zhou ; supervision, Zhansong Zhang and Chaomo Zhang; project administration, Zhansong Zhang and Chaomo Zhang. All authors have read and agreed to the published version of the manuscript.

References

- Zhang, S. N. (2008). Tight sandstone gas reservoirs: their origin and discussion. *Oil & Gas Geology*, 29 (1): 1-10,18.
- Chen, M., Dai, J. C., Liu, X. J., Qin, M. J., Pei, Y., & Wang, Z. T. (2018). Differences in the fluid characteristics between spontaneous imbibition and drainage in tight sandstone cores from nuclear magnetic resonance. *Energy & Fuels*, 32(10): 10333-10343.
- Diederix, K. M. (1982). Anomalous relationships between resistivity index and water saturation in the rotlligend sandstone. SPWLA 23th Annual Logging Symposium. Corpus Christi, Texas: Society of Petrophysicists and Well-Log Analysts.
- Archie, G. E. (1942). The electrical resistivity log as an aid in determining some reservoir characteristics. *Transactions of the AIME*, 146, 54-62.
- Lai J., Pang X., Xu F., Wang G., Fan X., Xie W., Chen J., Qin Z., Zhou Z. (2019). Origin and formation mechanisms of low oil saturation reservoirs in Nanpu Sag, Bohai Bay Basin, China. *Marine and Petroleum Geology* 110, 317–334.
- Hill, H. J., & Winsauer, J. D. (1956). Effect of clay and water salinity on electrochemical behaviour of reservoir rock. *AIME*, 207: 65-72.
- Clavier, C., Coates, G., & Dumanoir, J. (1977). Theory and experimental basis for the dual water model for interpretation of shaly sands. *SPE* 6859.
- Lima, O. A. L. (1995). Water saturation and permeability from resistivity, dielectric and porosity logs. *Geophysics*, 60(11):1756-1764.
- Zeng, W., C (1991). *New Technology of Oil-Gas Reservoir Logging Evaluation*. Beijing: Petroleum Industry Press.
- Berg, C. R. A. (1995). Simple effective-medium model for water saturation in porous rocks. *Geophysics*, 60(4): 1070-1080.
- Hu, S. F., ZHOU, C. C., Li, X., Li, C. L., & Zhang, S. Q. (2017). A tight sandstone trapezoidal pore oil saturation model. *Petroleum Exploration and Development*, 44 (5): 827-836.
- Li, X., Li, C. L., Li, B., Liu, X. F., & Yuan, C. (2020). Response laws of rock electrical property and saturation evaluation method of tight sandstone. *Petroleum Exploration and Development*, 47(1): 214-224.
- Hu, X. Y., Yuan, W., Tang, D., Cheng, R., & Yang, Y. (2018). Calculation method of shaly sands reservoir water saturation based on petrophysical facies. *Progress in Geophysics*, 33(02): 808-814.
- Zhang, W. Q., Song, X. M., Xiao, Y. X., Xu, F., & Hou, X. L. (2016). New method for initial oil saturation characterization of tuffaceous reservoir. *Well Logging Technology*, 40(03): 286-291.
- Herrick, D. C., & Kennedy, W. D. (1993). Electrical efficiency: a pore geometric model for the electrical properties of rocks. *Spwla Annual Logging Symposium*.
- Purcell, W. R. (1949). Capillary pressures - Their measurement using mercury and the calculation of permeability therefrom. *Journal of Petroleum Technology*, 1(02): 39-48.
- Swanson, B. F. (1981). A simple correlation between permeabilities and mercury capillary pressures. *JPT*, December, 2488–2504.

18. Katz, A. J., & Thompson, A. H. (1986). Quantitative prediction of permeability and electrical conductivity in porous rock. *Seg. Tech. Program Expand. Abstr.* 6–7.
19. Pittman, E. D. (1992). Relationship of porosity and permeability to various parameters derived from mercury injection-capillary pressure curves for sandstone. *AAPG Bull.* 76(2): 191-198.
20. Aguilera, R. (2004). Intergration of geology, petrophysics, and reservoir engineering for characterization of carbonate reservoirs through Pickett plots. *AAPG Bulletin*, 88 (4): 433-446.
21. Thomeer, J. H. M. (1960). Introduction of a pore geometrical factor defined by the capillary pressure curve. *Trans AIME*, 213: 354–358.
22. Skalinski, M. T., Kenter, J. A. M., & Jenkins, S. (2009). Rock type definition and pore type classification of a carbonate platform, Tengiz field, republic of Kazakhstan. *Society of Petrophysicists and Well-Log Analysts*, 1–18.
23. Farshi, M., Moussavi-Harami, R., Mahboubi, A., & Khanehbad, M. (2018). Reservoir rock typing using integrating geological and petrophysical properties for the Asmari Formation in the Gachsaran oil field, Zagros basin. *Journal of Petroleum Science and Engineering*, 176 (2019), 161-171.
24. Ferreira, F.C., Booth, R., Oliveira, R., Carneiro, G., Bize-Forest, N., & Wahanik, H. (2015). New rock-typing index based on hydraulic and electric tortuosity data for multi-scaledynamic characterization of complex carbonate reservoirs. SPE-175014-MS, SPE Annual Technical Conference and Exhibition, 28–30 September, Houston, Texas, USA.
25. Kolodzie, S. (1980). Analysis of pore throat size and use of the Waxman-Smiths equation to determine OOIP in Spindle field. In: Colorado. SPE-9382-MS, SPE Annual Technical Conference and Exhibition, 21–24 September, Dallas, Texas.
26. Clerke, E. A., Mueller III, H. W., Phillips, E. C., Eyvazzadeh, R. Y., Jones, D. H., Ramamoorthy, R., & Srivastava, A. (2008). Application of Thomeer Hyperbolas to decode the pore systems, facies and reservoir properties of the Upper Jurassic Arab D Limestone, Ghawar field, Saudi Arabia: a Rosetta Stone approach, *GeoArabia*, 13(4): 113–160.
27. Gomes, J. S., Ribeiro, T., Strohmenger, C. J., Negahban, S., & Kalam, M. Z. (2008). Carbonate reservoir rock typing-the link between geology and SCAL. *Society of Petroleum Engineers*, pp.15.
28. Ozkan, A., Cumella, S. P., Milliken, K. L., & Laubach, S. E. (2011). Prediction of lithofacies and reservoir quality using well logs, late cretaceous williams fork formation, mamm creek field, piceance basin, colorado. *Aapg Bulletin*, 95(10): 1699-1723.
29. Lai, J., Wang, G., Wang, S., Cao, J., Li, M., Pang, X., Zhou, Z., Fan, X., Dai, Q., Yang, L., He, Z., & Qin, Z. (2018). Review of diagenetic facies in tight sandstones: Diagenesis, diagenetic minerals, and prediction via well logs. *Earth-Science Reviews*, 185: 234-258.
30. Wang, J., Cao, Y., Liu, K., Liu, J., & Kashif, M. (2017). Identification of sedimentary-diagenetic facies and reservoir porosity and permeability prediction: an example from the eocene beach-bar sandstone in the dongying depression, china. *Marine & Petroleum Geology*, 82: 69-84.
31. Byrnes, A. P. (1997). Reservoir characteristics of low-permeability sandstones in the Rocky Mountains. *Mountain Geol.* 34: 39-48.
32. Ren, J. H., Zhang, L., Ezekiel, J., Ren, S. R., & Meng, S. Z. (2014). Reservoir characteristics and productivity analysis of tight sand gas in Upper Paleozoic Ordos Basin China. *J Nat Gas Sci Eng.* 19: 244-50.
33. Zhou, X. Q., Zhang, C., Zhang, Z. S., Zhang, R. F., Zhu, L. Q., & Zhang, C. M. (2019). A saturation evaluation method in tight gas sandstones based on diagenetic facies. *Marine and Petroleum Geology*, 107(2019): 310-325.
34. Zheng, D. Y., Pang, X. Q., Jiang, F. J., Liu, T. S., Shao, X. H., & Huyan, Y. Y. (2020). Characteristics and controlling factors of tight sandstone gas reservoirs in the Upper Paleozoic strata of Linxing area in the Ordos Basin, China. *Journal of Natural Gas Science and Engineering*, 75, 103-135.
35. Guo, H. L., Zhang, M. B., & Hu, Q. Y. (2000). China National Rock Thin Section Identification Standard of SY/T 5368–2000. *Petroleum Industry Press*, Beijing, China, pp 28.
36. Lai J., Fan X., Liu B., Pang X., Zhu S., Xie W., Wang G. (2020). Qualitative and quantitative prediction of diagenetic facies via well logs. *Marine and Petroleum Geology*, 120, 104486.

37. Zou, C. N., Tao, S. Z., Zhou, H., Zhang, X. X., Dong-Bo, H. E., & Zhou, C. M. (2008). Genesis, classification, and evaluation method of diagenetic facies. *Petroleum Exploration & Development*, 35(5): 526-5.
38. Zhao, Y. F., Yang, S. C., Ren, R., & Wang, Y. (2019). Diagenetic characteristics and evolution of tight quartz sandstones of Shangshihezi Formation in Gaoqing area. *Journal of China University of Petroleum (Edition of Natural Science)*, 2019, 43(6): 23-31.
39. Liu, H., Zhao, Y., Luo, Y., Chen, Z., & He, S. (2015). Diagenetic facies controls on pore structure and rock electrical parameters in tight gas sandstone. *Journal of Geophysics & Engineering*, 12(4): 587-600.
40. Givens, W. W. (1986). Formation resistivity index and related equations based upon a conductive rock matrix model (CRMM). 27th SPWLA Paper.
41. Rasmus, J. C. (1983). A variable cementation exponent, m , for fracture carbonates. *The Log Analyst*, 24 (6): 13-23.
42. Liu, H. P., Luo, Y., Zhao, Y. C., Chen, Z. Y., & Mu, G. D. (2017). Effects of diagenetic facies on rock electrical properties in tight gas sandstones. *Earth Science*, 42(4): 652-660.
43. Aguilera, M. S., & Aguilera, R. (2003). Improved models for petrophysical analysis of dual porosity reservoirs. *Petrophysics*, 44(1): 21-35.
44. Pan, B. Z., Zhang, L. H., & Shan, G. Y. (2006). Progress in porosity model for fractured and vuggy reservoirs. *Progress in Geophysics*, 21(4): 1232-1237.

Figure 1. Regional geological map.

Figure 2. Histogram of the relative contents of the detrital composition.

Figure 3. The diagenetic sequence and diagenetic stage of reservoirs in LX block.

Figure 4. The characteristics of photographs in LX block. a 100µm, (a) feldspar dissolved. (b) 400µm, the feldspar is dissolved to produce grid, sieve and palisade secondary pores. (c) 100µm, the chlorite film. (d) 100µm, the secondary growth of quartz. (e) 400µm, early siderite cementation. (f) 200µm, iron calcite cementation. (g) 100µm, kaolinite filled intergranular pores. (h) 10µm, scaly kaolinite filling intergranular pores.

Figure 5. Relationship between F and Por.

Figure 6. Relationship between R_t and S_w .

Figure 7. Relationship between cementation index and component maturity, clay, rmean.

Figure 8. Relationship between m and n in 3 type of diagenetic facies.

Figure 9. Distribution diagram of porosity and permeability of 3 type of diagenetic facies.

Figure 10. Cross plot of diagenetic facies by the mercury distortion (skp) and r35.

Figure 11. Cross plot by displacement pressure (P_d) and rmean of diagenetic facies.

Figure 12. Triangle lithology map in LX block.

Figure 13. Cross plot by S_p and the mercury distortion (s_{kp}) of 3 types diagenetic facies in LX block.

Figure 14. Schematic by diagenetic facies. (a) Construction facies. (b) Cementation facies. (c) Destruction facies

Figure 15. Relationship between cementation exponent and porosity.

Figure 16. The relationship between cementation exponent m and total porosity with different content of non-connected pores ($mb=2$).

Figure 17. The relationship between cementation exponent m and total porosity with different content of non-connected pores ($mb=1.5$).

Figure 18. The relationship between cementation exponent m and total porosity with different content of non-connected pores ($mb=1.8$).

Figure 19. The relationship between cementation exponent m and total porosity with different content of non-connected pores ($m = 2.5$).

Figure 20. The relationship between cementation exponent m and total porosity at different dissolution pore contents ($mb=2$).

Figure 21. The relationship between cementation exponent m and total porosity at different dissolution pore contents ($mb=1.8$).

Figure 22. The relationship between cementation exponent m and total porosity at different dissolution pore contents ($mb=2.5$).

Figure 23. The relationship between cementation exponent m and total porosity at different dissolution pore contents ($mb=3$).

Figure 24. Comparison between evaluation results and experimental results of parameter m

N O.	Depth	Diagenetic facies	Porosity	F	m	Matrix pore	Non-connecte d pore	Dissolutio n pore
	(m)		(%)		(a=1)	(%)	(%)	(%)
1519	Destruction facies	8.9	83.52	1.83	2.54	6.36	\
1707.62	Destruction facies	4.1	85.82	1.39	2.46	1.64	\
1794.98	Destruction facies	6.3	149.39	1.81	1.80	4.50	\
1956.39	Destruction facies	2.82	135.64	1.38	1.41	1.41	\
1956.99	Destruction facies	2.72	118.55	1.32	1.36	1.36	\
1955.39	Destruction facies	3.18	122.71	1.39	1.27	1.91	\
1854.88	Cementation facies	7.25	113.62	1.80	\	\	\
1647.38	Cementation facies	4.51	144.94	1.61	\	\	\
1217.24	Cementation facies	10.7	76.61	1.94	\	\	\
1791.67	Cementation facies	8.4	150.41	2.02	\	\	\
1646.42	Cementation facies	9.05	103.85	1.93	\	\	\
1470.8	Cementation facies	2.53	203.18	1.45	\	\	\
1550.83	Construction facies	9.6	133.40	2.09	5.76	\	3.84
1284.11	Construction facies	10.9	81.80	1.99	4.42	\	6.48
1552.69	Construction facies	16.4	75.55	2.39	10.58	\	5.82
1716.78	Construction facies	10.6	45.68	1.70	4.54	\	6.06
1285.51	Construction facies	8.76	68.02	1.73	4.17	\	4.59
1470.23	Construction facies	12.2	54.33	1.90	6.10	\	6.10

Appendix B

Dear Royal Society Open Science associate editor and three anonymous reviewers:

Hello! The serial number of my article is RSOS-202122.R1. The name of my manuscript is Comparative study and discussion of diagenetic facies and conductivity characteristics based on experiments. I am very grateful to the editors for their meticulous and patient handling, as well as the advice of the three reviewers. This submission made me realize the shortcomings of my own articles. The three reviewers gave their ideas from different angles. We have modified the article on both technical and language aspects after receiving the advices of the editor and two reviewers. We really appreciate them for improving our manuscript, as well as ourselves by learning from the advices. Please excuse us for not replying immediately since we spent time on modifying article structure and improving the verbalities. We used AJE to polish the article language, and the certificate of AJE is attached below. We listed the modifications based on issues proposed by the two experts. You will see that the article is much better. Thanks again to the editors and reviewers! Here is our response to the review comments:

Reviewer: 1

1. There are too many figures in the manuscript, not sure if these are necessary, does these add to the paper? For example, figure 1 is cited from the other manuscript, it is not necessary to display again.

Reply: I am very grateful to the reviewer for handling and affirming the manuscript. We checked the figure of the ms, deleted the original figure 1 and the original figure 3, checked the figure content in the ms, and modified the original figure 2, figure 4, figure 12 and figure 14. Finally, according to the adjustment of the content in ms, all the figures are readjusted and numbered. Thanks again.

2. Introduction: More recent references should be added.

Reply: We have accepted the suggestion and added more recent references. Added and replaced reference 1,2,3,4,8,9,10,11,17,22,23,31,36. Thanks a lot.

3. Figure 4 need scales.

Reply: We are very sorry for our negligence of scales in the original figure 4(h). Re-edited the picture carefully and added scales in the figure 4.

4. core casting thin section-thin section, please clarify.

Reply: We have checked through the ms, unify into core casting thin section. Thank you for your meticulousness and patience.

5. In part 4.1. Tight sandstones: texture, composition and diagenetic facies, use the word lithic fragments instead of cuttings.

Reply: Thank you very much for your advice. Using the specialized vocabulary rock fragments instead of cuttings. And checked the specialized vocabulary in ms. For example, modified metamorphic rock debris to metamorphic lithic fragments, modified sedimentary rock debris to sedimentary lithic fragments, modified siliceous cement to quartz cement, etc.

6. Cut down some redundant parts in Conclusion.

Reply: We have revised the Conclusion section according to your suggestion, redundant parts have been deleted in the ms. Thank you.

Reviewer: 2

1. In Figure 2 and text of manuscript, rock fragments should be used instead of cuttings.

Reply: First of all, I am very grateful to the reviewer for the advice raised. Using the specialized vocabulary rock fragments instead of cuttings. And checked the specialized vocabulary in ms. For example, modified metamorphic rock debris to metamorphic lithic fragments, modified sedimentary rock debris to sedimentary lithic fragments, etc.

2. For Figure 12, the amount of quartz, feldspar and rock fragments in each of the studied samples should be presented as a table.

Reply: We have accepted the suggestion and added the amount of rock compositions in table 1. In this way, the data used in the ms is fully displayed and publicly shared.

3. In Figure 14, rock fragments should be used instead of cuttings and quartz secondary overgrowth is correct.

Reply: Thanks a lot. We have modified the Figure 14, and checked through the ms, modified the error. Thank you for your meticulousness and patience.

4. In classification of sandstones, refer to the folk (1980) classification.

Reply: Thank you very much for your advice. According to folk (1980) classification, modified the quartz sandstone to quartz arenite, modified the feldspathic quartz sandstone to subarkose, modified the lithic quartz sandstone to sublitharenite, modified the feldspathic quartz sandstone to arkose in Figure 12.

5. In Figure 4e, the opaque minerals are not in the form of cement but in the form of grains?

Reply: In Figure 4e, it is the early siderite cement. The early cement is mainly micrite calcite, which usually does not replace the clastic particles, and often presents a basal or continuous crystal distribution, the particles do not contact each other, and the sandstone has a large inter-particle volume. In order to better illustrate, the explanation is added in title of Figure 4e. Thanks a lot for your advice.

6. How is iron calcite diagnosed in Figure 4f? If the staining method is used, it should be mentioned in the study method.

Reply: We are very sorry for our negligence of explaining the identification method of iron calcite in the study area. Yes, used the alizarin red staining to diagnose the iron calcite. Added the staining method in the title of Figure 4f, and also added in the part 3. Materials and Experiments. Thanks.

7. The paragenetic sequence of these sandstones must be re-edited in the text with the figure.

Reply: It is really true as reviewer suggested that the paragenetic sequence of these sandstones should be re-edited and specific. Considering that the study of paragenetic sequence actually requires full and detailed argumentation and explanation, it is necessary to add enough content to get the results of the paragenetic sequence shown in the figure 4. However, this part of the content is not much related to the main content of the manuscript. The manuscript just wants to show that most of the tight sandstones in the study area are in the middle diagenetic stage. This conclusion can be obtained by quoting the references. Therefore, in order to briefly describe the content of the manuscript and reduce unnecessary misunderstandings, absolutely delete this part of the content, citing relevant references to explain paragenetic sequence, thank you very much for your help in pointing out this problem, it has greatly helped the improvement of the manuscript, thank you.

Reviewer: 3

On behalf of my co-authors, thank you very much for giving us an opportunity to revise our manuscript. Thank you very much for your help in pointing out these problem, it has greatly helped the improvement of the manuscript. We have made a lot of revisions to this article, absolutely serious. You can see very clearly, we have checked through the ms, adjusted the structure of the article and revised the error. In order to reply the question more clearly, we have made reply and modification instructions after each annotation in the PDF. Thank you for your help again.

Appendix C

Dear Royal Society Open Science associate editor and anonymous reviewers:

Hello! The serial number of my article is RSOS-202122.R1. The name of my manuscript is Comparative study and discussion of diagenetic facies and conductivity characteristics based on experiments. I am very grateful to the editors for their meticulous and patient handling, as well as the advice of the reviewers. We have modified the article on both the quality of figures and language aspects after receiving the advices of the editor and reviewers. We spent time on improving the verbalities again. We used AJE to polish the article language, and the certificate of AJE is attached below. We listed the modifications based on issues proposed by the experts. Thanks again to the editors and reviewers! Here is our response to the review comments:

Reviewer: 4

1. The ms have been revised and i AGREE WITH THE replies The authors provided

Reply: I am very grateful to the reviewer for handling and affirming the manuscript. Thanks. .

2. Improve the English grammar

Reply: We spent time on improving the verbalities again. We used AJE to polish the article language, and the certificate of AJE is attached below. Thanks a lot.

3. remove some reduant parts in the revised manuscript

Reply: Yes, we have checked the ms carefully and modified the error. Thank you for your meticulousness and patience.

4. Improve the quality of some figures, for instance ,add the scale in the thin sectionsand sem images

Reply: Yes, re-edited the picture carefully and improved the quality of the figures. Thank you very much for your help in pointing out this problem, it has greatly helped the improvement of the manuscript, thank you.